
# GFIT3: A full physics retrieval algorithm for remote sensing of greenhouse gases in the presence of aerosols

Zhao-Cheng Zeng[1,2], Vijay Natraj[3], Feng Xu[4], Sihe Chen[2], Fang-Ying Gong[2], Thomas J. Pongetti[3], Keeyoon Sung[3], Geoffrey Toon[3], Stanley P. Sander[3], and Yuk L. Yung[2,3]

[1]Joint Institute for Regional Earth System Science & Engineering (JIFRESSE), University of California, Los Angeles, USA;
[2]Division of Geological and Planetary Sciences, California Institute of Technology, Pasadena, USA;
[3]Jet Propulsion Laboratory, California Institute of Technology, Pasadena, USA;
[4]School of Meteorology, The University of Oklahoma, Oklahoma, USA.

*Correspondence to*: Zhao-Cheng Zeng (zcz@gps.caltech.edu)

**Abstract.** Remote sensing of greenhouse gases (GHGs) in cities, where high GHG emissions are typically associated with heavy aerosol loading, is challenging due to retrieval uncertainties caused by imperfect characterization of scattering by aerosols. We investigate this problem by developing GFIT3, a full physics algorithm to retrieve GHGs ($CO_2$ and $CH_4$) by accounting for aerosol scattering effects in polluted urban atmospheres. In particular, the algorithm includes coarse (including sea salt and dust) and fine (including organic carbon, black carbon, and sulfate) mode aerosols in the radiative transfer model.

The performance of GFIT3 is assessed using high spectral resolution observations over the Los Angeles (LA) megacity made by the California Laboratory for Atmospheric Remote Sensing–Fourier Transform Spectrometer (CLARS–FTS). CLARS–FTS is located on Mt. Wilson, California, at 1.67 km a.s.l. overlooking the LA basin, and makes observations of reflected sunlight in the near-infrared spectral range. The first set of evaluations are performed by conducting retrieval experiments using synthetic spectra. We find that errors in the retrievals of column-averaged dry air mole fractions of $CO_2$ ($XCO_2$) and

$CH_4$ ($XCH_4$) due to uncertainties in the aerosol optical properties and atmospheric *a priori* profiles are less than 1% on average. This indicates that atmospheric scattering does not induce a large bias in the retrievals when the aerosols are properly characterised. The methodology is then further evaluated by comparing GHG retrievals using GFIT3 with those obtained from the CLARS-GFIT algorithm (used for currently operational CLARS retrievals) that does not account for aerosol scattering. We find a significant correlation between retrieval bias and aerosol optical depth (AOD). Comparison of GFIT3 AOD retrievals

with collocated ground-based observations from AERONET shows that the developed algorithm produces very accurate results, with biases in AOD estimates of about 0.02. Finally, we assess the uncertainty in the widely used tracer-tracer ratio method to obtain $CH_4$ emissions based on $CO_2$ emissions, and find that using the $CH_4/CO_2$ ratio effectively cancels out biases due to aerosol scattering. Overall, this study of applying GFIT3 to CLARS-FTS observations improves our understanding of the impact of aerosol scattering on the remote sensing of GHGs in polluted urban atmospheric environments. GHG retrievals

from CLARS-FTS are potentially complementary to existing ground-based and space-borne observations to monitor anthropogenic GHG fluxes in megacities.



# 1 Introduction

Remote sensing of greenhouse gases (GHGs) in cities provides abundant datasets for quantifying urban carbon sources and sinks, complementary to *in situ* ground-based measurements. However, large source regions such as megacities are also typically associated with heavy aerosol loading. Atmospheric aerosols modify the path of solar radiation and thereby introduce uncertainties in the retrieval of GHGs from reflected and scattered sunlight measurements. It has been suggested that imperfect characterization of aerosol optical and microphysical properties is a significant source of error for GHG retrievals (**Butz et al., 2009; O'Dell et al., 2011**). Many different full physics retrieval algorithms, which explicitly account for atmospheric absorption and scattering, and surface reflection, in the radiative transfer (RT) forward modelling, have been developed for spaceborne instruments for retrieving column-averaged dry air mole fractions of atmospheric carbon dioxide (XCO$_2$) and methane (XCH$_4$). Examples of these instruments include Orbiting Carbon Observatory-2 (OCO-2; **Boesch et al., 2011; O'Dell et al., 2018; Reuter et al., 2017**), the Greenhouse gases Observing SATellite (GOSAT; **Bril et al., 2007; Butz et al., 2011; Yoshida et al., 2013**), and TanSat (**Wang et al., 2020; Yang et al., 2020**). Full physics algorithms for retrieving GHGs explicitly fit aerosol optical and microphysical properties in order to minimize biases induced by scattering. Although the GHG retrievals show good agreement with ground-based Total Carbon Column Observing Network (TCCON) results, the retrieved aerosol optical depth (AOD) values have larger differences compared with collocated AERONET measurements (**Nelson et al., 2016**), probably due to limited information content for aerosols and interference from other factors. To minimize data uncertainty, many operational GHG retrieval algorithms filter out retrievals with optically thick aerosols. Observations from these spaceborne instruments are made in side scattering directions (scattering angles between ~90 and 150 degrees), where aerosol scattering effects are greatly reduced compared to the forward scattering direction. However, for an observatory targeting urban GHGs from other vantage points, such as the California Laboratory for Atmospheric Remote Sensing – Fourier Transform Spectrometer (CLARS–FTS) (**Fu et al., 2014**), which is located on the top of Mt. Wilson overlooking the Los Angeles (LA) Basin, the measurements are made in both backscattering and forward scattering directions (**Zeng et al., 2020a**). There has been very little prior work investigating impacts of aerosol scattering on GHG retrievals for a wide range of viewing geometries.

The main objective of this study is to demonstrate the performance of a full physics algorithm (hereafter referred to as **GFIT3**), which is an extension of the widely used GFIT model, to retrieve GHGs in polluted urban atmospheres from spectra of reflected solar radiation. GFIT is a state-of-the-art profile scaling algorithm to retrieve gas concentrations and related atmospheric and instrumental parameters from absorption spectra. It has been the primary retrieval algorithm for the Total Carbon Column Observing Network (TCCON) network (**Wunch et al., 2011**), which has been the benchmark for validating satellite-based trace gas observations. GFIT has also been used to analyse spectra from the MkIV balloon (e.g., **Sen et al., 1996**) and ATMOS (e.g., **Irion et al., 2002**), and is also a critical component of the currently operational CLARS-GFIT retrieval algorithm (**Fu et al., 2014**). GFIT2 (**Connor et al., 2016**) is an upgraded version of GFIT that enables retrieval of vertical profiles of trace gases. GFIT2 uses optimal estimation (**Rodgers, 2000**) as the inverse method to retrieve GHGs at



different altitudes. However, both GFIT and GFIT2 do not account for scattering from molecules and particulates in the atmosphere. Such contributions are negligible in the near-infrared domain for instruments that measure directly transmitted solar spectra, such as TCCON. However, for GHG retrievals based on reflected solar radiation measurements (e.g., CLARS–FTS, GOSAT, and OCO-2), the aerosol scattering effect is important and needs to be accurately modeled.

In this study, we specifically focus on measuring GHGs in the LA megacity, which is one of the most polluted cities,

and the second largest contributor to carbon emissions in the US, using observations from CLARS–FTS. We investigate the impacts of aerosol scattering on GHG retrievals using GFIT3 to jointly retrieve GHGs ($CO_2$ and $CH_4$) and AODs of coarse and fine mode aerosols. CLARS–FTS observes reflected sunlight in the near-infrared spectral range. CLARS–FTS observations provide a unique dataset to study the impact of aerosol scattering effect because of (1) the large viewing zenith angle (>80 degrees) and larger range of scattering angles compared to spaceborne instruments (**Zeng et al. 2020a**); and (2) the

longer light path in the planetary boundary layer (PBL) that is a consequence of the large viewing zenith angle. While a longer light path increases the sensitivity of the measurements to anthropogenic emissions from LA, it also makes the measurements susceptible to light path change due to aerosol particles in the PBL (**Zhang et al., 2017**). As a result, any effects from aerosol scattering on GHG remote sensing become amplified for CLARS–FTS due to its observation geometry. It is therefore very important to have proper aerosol models with accurate optical properties, including phase function and single scattering albedo,

in order to obtain accurate GHG retrievals.

Another scientifically unique feature of CLARS–FTS, among instruments that measure surface reflected sunlight, is that it uses the $O_2$ $^1\Delta$ band at 1.27 µm instead of the $O_2$ A band at 0.76 µm that is traditionally used by spaceborne instruments to constrain surface pressure and aerosols. Since the $^1\Delta$ band is closer to the $CO_2$ and $CH_4$ absorption bands around 1.6 µm, scattering effects in the GHG absorption bands are likely to be better constrained. Also, the spectroscopy of the $^1\Delta$ band is

more accurately known than that of the A band. The $^1\Delta$ band was not selected by current spaceborne instruments because of contamination from airglow emitted by oxygen molecules in the upper atmosphere. Recently, **Bertaux et al. (2020)** showed that the airglow contribution can be distinguished and separated from the $O_2$ absorption signal. Usage of the $^1\Delta$ band will be tested in the upcoming MicroCarb mission (**Bertaux et al., 2020**), the first such attempt for a spaceborne instrument. Outcomes from this study will shed light on the merits of using the $^1\Delta$ band for GHG remote sensing.

The paper is organized as follows. The CLARS–FTS instrument is introduced in Section 2. The GFIT3 retrieval algorithm is described in Section 3. In Section 4, we demonstrate retrieval experiments using synthetic spectra to evaluate GFIT3. Retrieval results for CLARS observations are presented in Section 5, followed by discussions and conclusions in Sections 6 and 7, respectively.



## 2 California Laboratory for Atmospheric Remote Sensing (CLARS)

### 2.1 CLARS-FTS

CLARS-FTS was designed and built at the Jet Propulsion Laboratory. It is optimized for reflected sunlight measurements with high spectral resolution in the near-infrared region (4,000–15,000 cm$^{-1}$). CLARS–FTS uses a pointing system to target a set of predefined surface reflection targets (**Fig. 1**) in the LA basin, as well as a local diffuse reflector (Spectralon) for measurements of the free tropospheric background (**Zeng et al., 2020b**). In the Los Angeles Basin Survey (LABS) operating mode, the

pointing system stares at each surface reflection target in the LA basin and records atmospheric absorption spectra using reflected sunlight as the light source. In the absence of aerosols, as shown in **Fig. 1(a)**, sunlight travels through the PBL twice with a defined path: once on the way to the surface target and a second time from the surface target to CLARS–FTS. The resulting light path through the PBL is greater than 5 km (see Table 1 in **Wong et al., 2015**), which is several times longer than other commonly used viewing geometries, e.g., observing the direct solar beam from the surface, or measurement of

surface-reflected sunlight from aircraft and spacecraft vantage points. In the presence of aerosols, the light path changes mainly due to aerosol scattering along the path from the basin to the mountain top. Examples of single and multiple scattering are demonstrated in Fig. 1(a). CLARS covers the whole basin every 1.5 to 2 hours. Depending on the season, the total number of observations within a single day ranges from 160 to 260, and the number of repeated scans of the whole basin is between five to eight times over the same timeframe. Additional details can be found in **Fu et al. (2014)**. **Fig. 1(b)** shows examples of the

observed radiance in the $O_2$ absorption band centred at 7885 cm$^{-1}$, the weak $CO_2$ absorption band (hereafter referred to as W$CO_2$) at 6220 cm$^{-1}$, the $CH_4$ absorption band at 6076 cm$^{-1}$, and the strong $CO_2$ absorption band (hereafter referred to as S$CO_2$) at 4852 cm$^{-1}$. The absolute radiance, which is needed to constrain the aerosol scattering and the surface reflectance, is derived by calibrating the raw spectral data of digital numbers. The calibration factor is derived by comparing the CLARS-FTS spectra with that of a collocated ASD Spectroradiometer. The signal to noise ratio (SNR) for the W$CO_2$ and $CH_4$ bands is about

300±80; for the $O_2$ and S$CO_2$ bands, the SNR is about 100–150 depending on the surface target.

### 2.2 Observation geometries

Compared to low earth orbit satellites such as OCO-2/3, observations from CLARS–FTS have a larger range of aerosol scattering angles, mainly due to the diurnal and seasonal change of incident solar geometry (**Zeng et al., 2020c**). **Fig. 2** shows the diurnal change of aerosol scattering angle for six selected surface reflection points. In the morning, the surface reflection

points to the west (West Pasadena and Santa Monica) have large scattering angles that gradually change to smaller scattering angles in the late afternoon. The opposite pattern of change can be observed at reflection points to the east (Santa Fe dam and Rancho Cucamonga). At reflection points to the south (Santa Anita and Long Beach), the changes are smaller than at other targets. These changes are a result of the fixed viewing geometry for each surface reflection target but varying solar geometry. A detailed description of the angular scattering effect can be found in **Zeng et al. (2020c)**. This large range of angles, from

forward scattering (<90 degrees) to backward scattering (>90 degrees), means that a majority of the change in aerosol scattering





comes from angular variations. This also indicates that the aerosol scattering phase function is a key parameter that needs to be accurately modelled in order to obtain high fidelity RT calculations.



## 3 GFIT3: A full physics approach for retrieving XCO₂ and XCH₄ from CLARS–FTS observations

GFIT3 incorporates the following four major components: (1) a pre-processing step using the CLARS-GFIT algorithm to generate gas absorption coefficients and other related parameters, as well as the $O_2$ slant column density (SCD) for excluding cloudy and heavy aerosol loading soundings; (2) a forward RT model (RTM) to generate synthetic spectra in order to simulate observed CLARS–FTS spectra; (3) an inverse model based on optimal estimation to update the surface and atmospheric state vector to minimize the difference between model and observation; and (4) a post-processing screening step to filter out bad

retrievals. The workflow chart is shown in **Fig. 3**.

### 3.1 Pre-processing using CLARS-GFIT

The objective of pre-processing is to identify measurements that are affected by clouds and/or heavy aerosol loading and to exclude them before the full physics retrieval. We employ CLARS–GFIT (**Fu et al, 2014**), which is a modified version of the GFIT program, to retrieve $O_2$ SCD using the same spectral bands and spectroscopic parameters used by TCCON. Aerosol

scattering is not considered in CLARS–GFIT; the ratio of retrieved $O_2$ SCD to calculated $O_2$ SCD estimated from surface pressure reanalysis data (National Center for Environmental Prediction (NCEP) reanalysis in this study), denoted by $O_2$ ratio, acts as a proxy (**Zeng et al., 2020c**). We filter out data with (1) $O_2$ ratio less than 0.85 (low clouds and high aerosol loading) and larger than 1.02 (high clouds); (2) SNR less than 100; (3) solar zenith angle (SZA) larger than 70 degrees; and (4) spectral fit error larger than one sigma above the mean. The gas absorption coefficients, *a priori* atmospheric profiles, and solar lines

processed by CLARS-GFIT will also be used in the forward RTM of GFIT3.

### 3.1.1 Calibrating O₂ absorption cross section

Analysis of the $O_2$ ratio under different aerosol conditions reveals a systematic bias (about 2%) between the retrieved $O_2$ SCD and that calculated using the NCEP reanalysis surface pressure, even in situations when the atmosphere is clear (**Appendix Fig. A1**). Such a bias in the $^1\Delta$ band has been reported by **Washenfelder et al. (2006)**, who found that for the TCCON spectra,

the retrieved column $O_2$ is consistently $2.27 \pm 0.25\%$ higher than the dry pressure column estimated from the surface pressure. This bias in TCCON retrievals is consistent with values for CLARS–GFIT retrievals. A similar systematic bias was found by **Butz et al. (2011)** for the $O_2$ A–band at 0.76 μm from satellite observations. These biases are most likely attributable to spectroscopic uncertainties. We adopt a simple method of scaling the absorption cross sections in the $^1\Delta$ band by a factor of 1.02 to make our modelled radiances in the $^1\Delta$ band consistent with observations.

### 3.2 Forward model

### 3.2.1 Optical property-based principal component analysis RTM





RT models simulate the radiance based on inputs of the state vector and related model parameters. In theory, a sophisticated line-by-line RTM (e.g., LIDORT; **Spurr et al., 2008**) with a high number of computational quadrature angles (streams) is needed to accurately simulate the propagation of sunlight through the atmosphere. However, simulation of high resolution

CLARS–FTS spectra that require resolving gas absorption lines with fine spectral sampling is computationally expensive. Instead, many fast RTMs (e.g., **Butz et al., 2011; O'Dell et al., 2012; Somkuti et al., 2017**) have been developed to speed up the radiance calculation without introducing large systematic errors in the trace gas retrieval. In this study, we adopt an optical property-based principal component analysis (O-PCA) RTM developed by **Natraj et al. (2005, 2010)** and improved by **Kopparla et al. (2016, 2017)**. The O-PCA procedure was linearized and analytic Jacobians developed for the PCA-based

radiation fields by **Spurr et al. (2013)**. It has been shown to be fast and accurate for retrieving $CO_2$ from satellite measurements (**Somkuti et al., 2017**). The O-PCA method first divides the spectral region into bins. Each bin is characterized by grouping certain optical properties (such as atmospheric layer trace gas optical depth values or single scattering albedos) that are similar within the bin. The selection for spectral binning is typically based on the division of (the logarithms of) the total-atmosphere gas optical depths into decadal intervals. We use 11 bins in this study. For each bin, PCA is implemented on a dataset that

includes the extinction optical depth and single scattering albedo profiles, as well as the (wavelength-dependent) surface albedo and column optical depth for each aerosol type. High-accuracy line-by-line multiple scattering calculations (using LIDORT in this work) are then performed for profiles representing the bin mean and PCA-perturbed properties. For this analysis, we use 32 streams for these calculations. The multiple scattering calculations are computationally expensive; reduction of the number of these calculations is the main reason for the speed-up afforded by O-PCA. O-PCA also performs a fast and low-accuracy

line-by-line calculation of the radiances using the 2-Stream Exact Single Scattering (2S-ESS; **Spurr and Natraj, 2011**) model for every spectral point in the band. The 2S-ESS model computes both the single scattering contribution to the radiance and a two-stream approximation to the multiple scattering contribution. Finally, the total radiance field is obtained for every point in the bin by calculating a wavelength-dependent correction factor to adjust the 2S-ESS calculations. A detailed description of the O-PCA methodology can be found in **Kopparla et al. (2017)** and **Spurr et al. (2013)**. Simulations (see **Fig. 4**) show that,

while the accuracy of O-PCA depends on the aerosol loading, almost all of the spectral calculations have an error less than 0.1%. The root mean square error (RMSE) is less than 0.01%.


### 3.2.2 State vector

The state vector includes all variables that are to be retrieved by GFIT3 in order to fit the observed spectra. These variables
are inputs to the forward RTM. **Table 1** summarizes all the variables in the state vector and the values used for their
uncertainties in their retrieval.

**Table I. Summary of variables in the state vector and their uncertainties.**

| Variables | # of variables | A priori value | A priori uncertainty | Descriptions |
|---|---|---|---|---|
| $CO_2$ scale factor | 1 | 1.0 | 0.05 | *a priori* profile from CarbonTracker model |
| $CH_4$ scale factor | 1 | 1.0 | 0.05 | *a priori* profile constructed from GFIT and ground observations |
| $H_2O$ scale factor | 1 | 1.0 | 0.40 | *a priori* profile from NCEP |
| Surface pressure | 1 | NCEP | 2 hPa | |
| Surface albedo | 4 | Zeng et al., (2018) | 0.10, 0.07, 0.07, 0.04 | For the four bands: $O_2$, $WCO_2$, $CH_4$, and $SCO_2$ |
| Spectral continuum | 5×4 | 0 | 0.01, 0.005, 0.002, 0.0016, 0.001 | Zeroth to fourth orders of Legendre polynomial |
| Frequency shift | 4 | 0 | 0.1 | For the four bands |
| AOD coarse | 1 | 0.02 | 0.02 | Optical properties from GOCART |
| AOD fine | 1 | 0.01 | 0.02 | Optical properties from GOCART |
| Aerosol Layer Height | 1 | 0.70 | 0.05 | |
| Interference gas scale factors | 2 (HDO, $^{13}CO_2$) | 1.0 | 0.4, 0.02 | *a priori* profiles from GFIT |

### (1) $CO_2$ and $CH_4$ profiles

We follow the TCCON methodology and perform a retrieval that scales predefined vertical shapes of $CO_2$ and $CH_4$ to obtain
$XCO_2$ and $XCH_4$. This is faster and simpler than a full profile retrieval that independently scales gas mixing ratios at different
altitudes. The profile scaling method is also less sensitive to systematic errors related to the shape of the calculated spectral
lines, such as ILS and spectroscopic line widths (**Wunch et al., 2011**). Although a profile retrieval is possible, there are not
enough degrees of freedom in the measurement to fully resolve the gas profile. Therefore, the retrieval problem will be ill-
posed and under-determined if strong constraints are not imposed on the vertical profile. Sensitivity tests show that the profile
scaling approach is efficient and that errors from possible bias in the profiles are small (**Section 4**).

To account for GHG enhancement in the LA PBL, we used $CO_2$ simulations from the widely used CarbonTracker–
$CO_2$ model (**Peters et al., 2007**), which is an assimilation model incorporating available observations. 3-hourly simulations



are available from the CarbonTracker–$CO_2$ model. Monthly averaged $CO_2$ profiles are used as the *a priori* profiles in GFIT3

(**Fig. 5(a)**). For $CH_4$, since high resolution simulations are not available at city scale, we reconstruct the profiles based on CLARS-GFIT *a priori*. A constant PBL enhancement of 91 ppb, as estimated by **Verhulst et al. (2017; Table 5)** using the NASA megacity network, is added to the monthly averaged $CH_4$ profiles, as shown in **Fig. 5(b)**. Diurnal changes in the PBL enhancement are not considered in this analysis.

**(2) Surface albedo and aerosol properties**

The contributions to the observed radiance from surface reflectance and aerosol scattering are coupled. Similar to **Zeng et al. (2018)**, we assume a Lambertian surface and calculate the *a priori* surface albedo by ratioing the measured radiance reflected from the surface target to that reflected by a Spectralon board beside the FTS. The Spectralon measurement represents the incident radiance before entering the PBL. For aerosols, we use AOD values from MERRAero reanalysis data (**Rienecker et al., 2011**) and associated optical properties from GOCART (**Chin et al., 2002**), which includes five aerosol types: sea salt,

dust, organic carbon, black carbon, and sulfate. In light of the difficulty in resolving so many aerosol types from measurements, we separate the five aerosol types into two groups based on size: coarse mode (sea salt and dust) and fine mode (organic carbon, black carbon, and sulfate). While the sizes, extinction efficiency, and phase function of aerosols in the fine mode are similar, the black carbon has a much smaller single scattering albedo (SSA). For sea salt and dust aerosols, five differently-sized bins are separately tracked in the MERRA model. The sea salt, black carbon, organic carbon, and sulfate are all

hygroscopic. GFIT3 uses monthly average aerosol optical properties (extinction efficiency, SSA, and phase function) at four daytime hours (local times 7h,10h,13h,16h). The monthly averaged density fraction of aerosols is shown in **Fig. 6**. While the fine mode aerosols show identical monthly variabilities, the coarse mode particles show a clear seasonal cycle, with more sea salt in summer originating from the ocean and more dust in winter originating from the Mojave desert and transported to the LA basin. **Fig. 7** shows the wavelength dependence of aerosol optical properties averaged over all months in 2013. Fine mode

aerosols have a larger Angstrom exponent, and hence a greater wavelength dependence, than coarse mode aerosols.

In the retrieval algorithm, we retrieve AODs for the coarse mode and fine mode, and the aerosol layer height (ALH; **Table 1**). The geometric thickness of the aerosol layer is not considered here since the measurement is not sensitive to that quantity (**Zeng et al., 2020a**). The *a priori* AODs are derived from monthly averaged AERONET observations at Caltech, and the *a priori* ALH from an aerosol profiling lidar (MiniMPL), also at Caltech (**Zeng et al., 2018**).

**(3) Surface pressure**

The *a priori* surface pressure is extracted from NCEP reanalysis data (**Kalnay et al., 1996**), which is used for GGG2014 TCCON retrievals (**Wunch et al.,** 2015). A comparison with ECMWF ERA5 reanalysis (**Hersbach et al., 2020**), which has a higher resolution, indicates that the two surface pressure datasets are highly correlated, with a standard deviation of the difference of about 2 hPa (**Zeng et al., 2020b**). In the GFIT3 retrieval, we assume this value as the uncertainty for surface

pressure.





### 3.2.3 Solar model

To construct the high-resolution solar irradiance, we combine the solar continuum level estimated from the solar spectrum

developed by **Kurucz (2005)** (http://kurucz.harvard.edu/sun/irradiance2008/) and the high resolution solar pseudo-transmittance spectrum from GFIT **(Toon, 2014;** https://mark4sun.jpl.nasa.gov/toon/solar/solar_spectrum.html**)**. The Kurucz spectrum was created from the solar spectrum measured by a high-resolution FTS at the Kitt Peak National Observatory. In the near infrared spectral regions of relevance to this work, Toon's solar pseudo-transmittance spectrum is a combination of high-resolution spectra from balloon FTS, ground-based Kitt Peak and TCCON observations. A similar combination of Kurucz

and Toon reference spectra was also used by GOSAT (**Yoshida et al., 2013**). The absolute solar irradiance is necessary to constrain aerosol scattering and surface reflectance.

### 3.2.4 Jacobian

The Jacobian matrix contains the first order derivative of the simulated radiance with respect to all state vector elements, and is a key variable in inverse modeling to fit the observed spectra by iteratively optimizing the state vector. This matrix has a

dimensionality of $m \times n$, where $m$ refers to the number of measurement channels and $n$ is the number of state vector elements. **Fig. 8** illustrates a sample Jacobian matrix calculated by O-PCA.

## 3.3 Inverse modeling

### 3.3.1 Optimal estimation

Mathematically, the measurement vector $y$, which is the observed CLARS–FTS radiance, is related to the state vector $x$,

including $O_2$, $CO_2$, and $CH_4$ SCDs, and other relevant geophysical parameters, through a forward model $\mathbf{F}$ and model parameter vector $b$:

$$y = \mathbf{F}(x, b) + \varepsilon \qquad (1)$$

Specifically, $b$ is a set of input parameters for the forward model that are not retrieved, such as gas absorption coefficients and observing and solar geometries, while the state vector $\mathbf{x}$ is a set of parameters to be retrieved, such as trace gas columns, aerosol

properties, and surface properties. The forward model $\mathbf{F}$ is a RT model (O-PCA in this study) that simulates the radiance based on input parameters $b$ and $x$. $\varepsilon$ is the error vector containing both the measurement noise and the forward model error. The goal of optimal estimation is to produce a maximum *a posteriori* estimate of the state vector that minimize the following cost function **(Rodgers, 2000)**:

$$J(x) = \chi^2 = [y - \mathbf{F}(x, b)]^T \mathbf{S}_\varepsilon^{-1} [y - \mathbf{F}(x, b)] + (x - x_a)^T \mathbf{S}_a^{-1} (x - x_a) \qquad (2)$$

where $x_a$ is the *a priori* state vector; $\mathbf{S}_a$ is the *a priori* covariance matrix for the state vector; and $\mathbf{S}_\varepsilon$ is the measurement error covariance matrix. In this study, the measured radiance from the $O_2$ $^1\Delta$, WCO2, CH4, and SCO2 absorption bands constitutes the measurement vector $\mathbf{y}$. For $\mathbf{S}_\varepsilon$, we assume that there is no cross-correlation between different spectral channels, resulting in a diagonal matrix. The spectral error term $\varepsilon$ includes the measurement noise, which can be characterized by SNR, and uncertainty in the forward model. To estimate forward model uncertainty, we use the results from **Figure 4**, which are spectral





fitting error estimates between O-PCA and LIDORT. The RMSE is less than 0.01%, which is much smaller than the

measurement noise. We therefore use the measurement noise to generate the matrix $\mathbf{S}_\varepsilon$. We adopt the Levenberg-Marquardt

method (**Levenberg, 1944; Marquardt, 1963; Rodgers, 2000**) to obtain the optimal estimate of the state vector $\boldsymbol{x}$ that

minimizes the cost function $J(\boldsymbol{x})$ through an iterative process:

$$\boldsymbol{x}_{i+1} = \boldsymbol{x}_i + [(1+\gamma)\mathbf{S}_a^{-1} + \mathbf{K}_i^T\mathbf{S}_\varepsilon^{-1}\mathbf{K}_i]^{-1}\{\mathbf{K}_i^T\mathbf{S}_\varepsilon^{-1}[\boldsymbol{y} - \mathbf{F}(\boldsymbol{x}_i, \boldsymbol{b})] - \mathbf{S}_a^{-1}[\boldsymbol{x}_i - \boldsymbol{x}_a]\} \tag{3}$$

where the subscript $i$ indicates the $i^{th}$ iteration; The parameter $\gamma$ is chosen at every step to minimize the cost function. Initially

it is set to be 10; $\mathbf{K}$ is the Jacobian matrix, which is the first derivative of $\boldsymbol{F}(\boldsymbol{x}, \boldsymbol{b})$ with respect to $\boldsymbol{x}$:

$$\mathbf{K}_i = \partial\mathbf{F}(\boldsymbol{x}_i, \boldsymbol{b})/\partial\boldsymbol{x}_i \tag{4}$$

where each element in $\mathbf{K}_i$ defines the sensitivity of the simulated radiance to the corresponding geophysical variable in the

state vector. At each step, the parameter $\gamma$ is updated based on the ratio $R$ (**Fletcher, 1971**):

$$R = (\chi_i^2 - \chi_{i+1,true}^2)/(\chi_i^2 - \chi_{i+1,forecast}^2) \tag{5}$$

where $\chi_{i+1,true}^2$ refers to the cost function computed with the updated state vector $\boldsymbol{x}_{i+1}$ in the forward model $\mathbf{F}_{i+1} = \mathbf{F}(\boldsymbol{x}_{i+1}, \boldsymbol{b})$,

while $\chi_{i+1,forecast}^2$ is computed using a linear approximation to the forward model $\mathbf{F}_{i+1} = \mathbf{F}_i + \mathbf{K}_i * (\boldsymbol{x}_{i+1} - \boldsymbol{x}_i)$. $R$ quantifies

the impact of forward model nonlinearity on cost function reduction. If the linear approximation is perfect, then $R$ will be unity

since $\chi_{i+1,true}^2 = \chi_{i+1,forecast}^2$. The strategy for updating $R$ is as follows: if $R$ is greater than 0.75, then reduce $R$ by a factor of

2; if R is less than 0.25, then increase $R$ by a factor of 10; otherwise, leave $R$ unchanged. Convergence is achieved when the

change in the state vector is small compared to the *a posteriori* error:

$$d_i^2 = (\boldsymbol{x}_i - \boldsymbol{x}_{i+1})^T\hat{\mathbf{S}}^{-1}(\boldsymbol{x}_i - \boldsymbol{x}_{i+1}) \ll n \tag{6}$$

where $n$ is the number of state vector elements; $\hat{\mathbf{S}}^{-1}$ is the *a posteriori* error covariance matrix for the estimated state vector

$\hat{\boldsymbol{x}}$. At convergence, $\hat{\mathbf{S}}^{-1}$ can be estimated as follows:

$$\hat{\mathbf{S}} = (\mathbf{K}^T\mathbf{S}_\varepsilon^{-1}\mathbf{K} + \mathbf{S}_a^{-1})^{-1} \tag{7}$$

where $\hat{\mathbf{S}}$ includes the *a posteriori* uncertainties of all retrieved elements in the state vector and their correlations.

### 3.3.2 Averaging kernel

Similar to TCCON, we use the column averaging kernel calculated from our retrieval algorithm to quantify the altitude-

dependent sensitivity of the total column retrievals to changes in the vertical profile of partial column densities. Ideally, the

column averaging kernel would be unity at all altitudes, meaning a unit change in partial column at any altitude would lead to

the same amount of change in the total column. In practice, however, the column averaging kernel is not a perfect unity vector.

To derive the column averaging kernel, we first calculate the full averaging kernel matrix ($m \times m$):

$$\mathbf{A} = (\mathbf{K}^T\mathbf{S}_\varepsilon^{-1}\mathbf{K} + \mathbf{S}_a^{-1})^{-1}\mathbf{K}^T\mathbf{S}_\varepsilon^{-1}\mathbf{K} \tag{8}$$

where $m$ is the number of atmospheric layers. $A_{ij}$ represents the derivative of the retrieved mixing ratio at level $i$ with respect

to the true mixing ratio at level $j$. The $j^{th}$ element of the column averaging kernel is given by:


$$a_j = \sum_i A_{ij} \frac{\Delta p_i}{\Delta p_j} \tag{9}$$

where $\Delta p_i$ is the pressure thickness at level $i$. $a_j$ describes the change in the retrieved total column abundance with respect to a perturbation of the partial column at the $j^{th}$ atmospheric level. **Fig. 9** shows examples of column averaging kernels for $CO_2$

and $CH_4$ at different SZA values. Both spectral channels show similar shape and have higher averaging kernel values (close to 1) in the troposphere than in the stratosphere. For comparison of CLARS-FTS measurements with other datasets (such as satellite observations), the above averaging kernels and *a priori* profiles from CLARS-GFIT should be taken into account. Details about implementation of the averaging kernel correction can be found in **Wunch et al. (2011)**.

### 3.4 Post-processing

After obtaining the SCDs for $O_2$, $CO_2$, and $CH_4$, $XCO_2$ and $XCH_4$ can be calculated as follows:

$$XCO_2 = \frac{CO_2\ SCD}{O_2\ SCD} \times 0.2095 \tag{10}$$

$$XCH_4 = \frac{CH_4\ SCD}{O_2\ SCD} \times 0.2095 \tag{11}$$

where the constant 0.2095 is the column-averaged dry-air mixing ratio of $O_2$ in the atmosphere. In the post-processing, multiple filters are applied to ensure good retrieval quality. First, retrievals that fail to converge after 15 iterations according to the

procedure outlined in **Equation (6)** are excluded. Second, the spectral fitting residual (RMSE) for each window should be smaller than 0.01 for all four bands. Third, outliers in retrieved state vector parameters, including $O_2$, $CO_2$, and $CH_4$ SCDs, that have large impact on $XCO_2$ and $XCH_4$, are filtered. In this study, we define outliers as values that are more than three standard deviations away from the mean. For retrievals of CLARS–FTS observations from June 2013 to May 2014, about 80% of all pre-filtered observations pass the post-processing filters.




## 4. Inversion experiments based on synthetic spectra

The goal of applying the GFIT3 algorithm to simulated synthetic spectra is to assess the performance of the algorithm in retrieving $XCO_2$ and $XCH_4$ and to quantify the impacts on the accuracy due to factors such as aerosol scattering, imperfect meteorological data, RTM errors, uncertainty in gas absorption, and instrument noise. In this study, we primarily concentrate
on three potentially important error sources: imperfect characterization of aerosol scattering, assumptions about the vertical distributions of $CO_2$ and $CH_4$, and biases due to usage of the O-PCA RTM.

We first generate synthetic spectra using LIDORT with high accuracy (32 streams) to reproduce the "true" spectra under three aerosol loading scenarios (total column AOD = 0.01, 0.05, and 0.1), which covers the AOD range for non-cloudy days based on AERONET-Caltech measurements (**Appendix Fig. A2**). Given that CLARS–FTS observes large air mass
factors (more than eight times the vertical column) in the PBL because of the long slant column in the line of sight, the aerosol loading along the slant path is much higher than the column AOD. To simulate the synthetic spectra, we use 3-hourly aerosol composition data from MERRA aerosol reanalysis and other optical properties (SSA and phase function) from the GOCART model (**Section 3.2.2**). $CO_2$ and $CH_4$ vertical profiles are derived as described in **Section 3.2.2**. The hourly variability of $CH_4$ in the PBL is assumed to be the same as that of $CO_2$ since they are co-emitted and follow a similar atmospheric mixing process.
Surface albedos for the $O_2$, $WCO_2$, $CH_4$, and $SCO_2$ bands are estimated from CLARS–FTS observations. All other inputs are the same as the state vector described in **Section 3.2.2**. Measurement noise (which we assume to be white noise with a mean of 0 and a standard deviation of 1/SNR) is added to generate the synthetic spectra as a proxy for CLARS–FTS observations. We test the GFIT3 algorithm on the synthetic spectra for the three surface targets at Santa Anita, Santa Fe, and West Pasadena over a wide range of observing geometries encompassing four seasons (Jan, April, July, and October) and five hours from
early morning to late afternoon (~8–9h, 10–11h, 12–13h, 14–15h, 16–17h). Since data in the early morning and late afternoon hours may not be available in winter, we select observations from the available day time data with a time step of at least one hour. In total, 60 different observation scenarios are selected.

We conduct four retrieval tests on the synthetic spectra, as listed in **Table 2**. In Test 1, we assume perfect knowledge of aerosol composition and GHG profiles. The goal is to assess the capability of O-PCA and the inverse framework for
retrieving $XCO_2$ and $XCH_4$. In Test 2, we use O-PCA, but with monthly average aerosol composition and GHG profiles. The goal is to investigate retrieval uncertainty due to assumptions about aerosols and GHG profiles, and RT calculation approximations. Test 3 is similar to Test 2, except that we use 3-hourly GHG profiles. The goal is to isolate the impact of uncertainty in aerosol composition. Test 4 is also similar to Test 2, except that we use 3-hourly aerosol composition. The goal is to isolate the impact of imperfect knowledge of GHG vertical distribution.
**Fig. 10** shows results from Test 1. It is evident that all simulations have mean absolute error (MAE) less than 0.5%. The retrieval error, however, increases as AOD increases. In the haziest scenario (AOD = 0.1), the largest retrieval error is around 1%. Results from Test 2 (**Fig. 11**) are broadly similar to those from Test 1. The errors are generally larger than those in Test 1 due to the bias in aerosol optical properties and atmospheric profiles. On average, the MAEs are less than 1%; the



largest errors are greater than 2%. The bias in the retrieved AOD is smaller at larger AOD values because of the stronger
aerosol scattering signal. Moreover, the bias in ALH is about -10% on average, indicating an average error less than 1 km. **Fig.
12** shows results from Tests 3 and 4. No clear correlation can be observed between bias in $XCO_2$ and $XCH_4$ retrievals and that
in aerosol optical properties for either coarse or fine mode aerosols. This indicates that a combination of fine and coarse mode
aerosols is able to accurately capture the scattering effects. On the other hand, there is a clear correlation between bias in the
trace gas columns and that in PBL enhancement. However, the MAE is still almost always less than 1%.

**Table 2. Synthetic experiments to assess the impact of RTM, aerosol composition, and GHG profiles on
retrievals of $XCO_2$ and $XCH_4$ from CLARS-FTS observations.**

| Experiment | Aerosol composition | Atmospheric profile | Radiative transfer model | Objective |
|---|---|---|---|---|
| Synthetic spectra | 3-Hourly | 3-Hourly | LIDORT | To create synthetic spectra |
| **Test 1**: Noise free simulation | 3-Hourly | 3-Hourly | O-PCA | To investigate the error due to RTM approximations |
| **Test 2**: Operational algorithm | Monthly | Monthly | O-PCA | To investigate the error due to the operational algorithm |
| **Test 3**: Aerosol impact | Monthly | 3-Hourly | O-PCA | To investigate the error due to assumptions about aerosol composition |
| **Test 4**: Vertical profile impact | 3-Hourly | Monthly | O-PCA | To investigate the error due to assumptions about vertical distribution of $CO_2$ and $CH_4$ |

## 5. Retrieval results for CLARS–FTS observations

We applied the GFIT3 retrieval algorithm to one-year of CLARS observations from June 2013 to May 2014. Over this period,
CLARS–FTS spent a large portion of measurement time observing the Santa Anita, Santa Fe, and West Pasadena targets.
Therefore, these three surface reflection points are our focus in this section. In total there are 36,170 observed spectra from
CLARS–GFIT. After pre-processing, we obtain 12,911 spectra that pass the filters for processing by the GFIT3 algorithm.
Most of the retrievals converge after less than 10 iterations. However, about 20% of the measurements fail to pass the post-
processing filters and are not included in this analysis.

### 5.1 Residuals from spectral fitting

**Fig. 13** shows normalized residuals with respect to the continuum level from spectral fitting for the $O_2$, $WCO_2$, $CH_4$, and $SCO_2$
bands. The RMSE values are less than 1%, and the majority of residuals are less than 0.5%. The $SCO_2$ band shows a larger
residual compared to the other bands, partly due to imperfect spectroscopic data (**Crisp et al. 2012**), and partly due to the large
aerosol scattering contribution, especially in the strong absorption lines (of which there are several due to the high spectral
resolution of CLARS–FTS). It is instructive to compare these results with fitting residuals from CLARS-GFIT (**Appendix
Fig. A3**), where aerosol scattering is neglected. It is evident that the residuals from GFIT3, especially in the $SCO_2$ band, are





significantly smaller. In the GFIT3 algorithm, the aerosols are primarily constrained by the $O_2$ and the $SCO_2$ bands. This is because *a priori* for atmospheric pressure are very accurate (~0.2% uncertainty) and $O_2$ concentration well known, thereby resulting in the $O_2$ absorption spectra providing strong constraints on the aerosol scattering effects. For the $SCO_2$ band, since most of the absorption lines are saturated, any extra radiance in this spectral region is attributable to aerosol scattering. Results

from this study suggest that the effects of scattering in the observed spectra can be accurately characterized by the aerosol models used in the GFIT3 algorithm. Not accounting for scattering leads to large spectral fitting residuals, and thereby large biases, in GHG retrievals.

## 5.2 Retrievals of $XCO_2$ and $XCH_4$

**Fig. 14** compares $XCO_2$ and $XCH_4$ retrievals from GFIT3 (after post-processing) and CLARS-GFIT. In general, when aerosols

are not accounted for in the retrieval, as in CLARS-GFIT, $XCO_2$ and $XCH_4$ are overestimated (see discussion in **Section 6.2**). The bias can be up to about 10% for both $XCO_2$ and $XCH_4$. The scatter plots indicate that the differences in $XCO_2$ and $XCH_4$ are significantly correlated with AOD. The correlation coefficients are higher for Santa Anita, probably because of the smaller changes in scattering angle (and therefore aerosol effects) compared to the other two surface targets. The $XCO_2$ and $XCH_4$ differences are mostly 10–30 ppm and 50–150 ppb, respectively, for an AOD value of 0.05 in the $^1\Delta$ absorption band. Since

$CO_2$ and $CH_4$ are retrieved at similar wavelengths, the biases in $XCO_2$ and $XCH_4$ retrievals due to aerosol scattering are expected to be comparable. The impact on the retrieved $XCH_4/XCO_2$ ratio in the presence of aerosols is further discussed in **Section 6.1**. **Figure A4** shows comparisons for all 28 surface targets based on available measurements from June 2013 to May 2014.

## 5.3 Comparison of retrieved AOD with AERONET and ALH with MiniMPL

We compare the retrieved AOD with ground-based AERONET observations at Caltech. AERONET is a global ground-based aerosol monitoring network (**Holben et al., 1998**) that has been providing high-accuracy measurements of total AOD from the ultraviolet to the near infrared. The AERONET instrument at Caltech is located on the university campus in Pasadena, which is geographically close to the Santa Anita, Santa Fe, and West Pasadena surface targets. The AERONET-Caltech measurements cover the wavelength range from 340–1020 nm. To derive the AOD in the $O_2$ $^1\Delta$ band, we extrapolate from

AERONET measurements using the Ångström exponent law (**Seinfeld and Pandis, 2006**). **Fig. 15** shows that the retrieved AOD is in good agreement with AERONET-Caltech AOD, with RMSE values of about 0.02. The difference is larger for higher AOD. This effect may be due to two reasons. First, CLARS-FP retrievals have higher uncertainty at large AOD values because of the magnification of biases due to misrepresentation of aerosol optical properties. Second, CLARS-FTS and AERONET observe different parts of the atmosphere due to differences in their observing geometries. Considering the spatial

heterogeneity of aerosol distribution, such difference between retrieved and observed AOD is expected. The retrieved ALH values agree closely with MiniMPL observations (**Fig. 16**); however, they do not have significant correlation on a point-by-point basis (not shown). This suggests the difficulty in constraining ALH when it is jointly retrieved with GHGs. The signal



from ALH may be interfered with by imperfect characterization of other factors that existing full physics algorithms cannot resolve. However, when ALH is retrieved independently using specifically targeted $O_2$ absorption lines, high accuracy can be

achieved (**Zeng et al., 2019**).


## 6. Discussions

### 6.1 Testing the assumption that the ratio between XCH$_4$ and XCO$_2$ is not affected by aerosol scattering

The tracer-tracer ratio method to retrieve CH$_4$ emissions based on CO$_2$ emissions or CH$_4$ concentration based on CO$_2$
concentration is based on the assumption that the CH$_4$/CO$_2$ ratio cancels out any systematic errors caused by aerosol scattering
in the two bands (e.g., **Frankenberg et al., 2005; Parker et al., 2011; Wong et al., 2015, 2016; He et al., 2019**). However,
the fact that the spectral regions do not exactly overlap and that the line intensities have different strengths may reduce the
validity of this assumption. Since XCO$_2$ and XCH$_4$ are simultaneously retrieved from both GFIT3 and CLARS-GFIT, these
retrievals serve as good datasets for testing the ratioing assumption. **Fig. 17** shows a scatter plot between CLARS-GFIT
XCH$_4$/XCO$_2$ ratios and those from GFIT3. No systematic bias is observed from this comparison, suggesting high accuracy of
using the trace-tracer ratio method to accurately estimate CH$_4$ emissions using remote sensing measurements in the presence
of aerosols.

### 6.2 Impact of aerosol scattering on XCO$_2$ and XCH$_4$ retrievals regulated by surface reflectance

The effects of aerosol scattering and surface reflectance on modifying the path of solar radiation, and thereby introducing
biases in trace gas retrievals, are coupled. A darker surface means a relatively higher contribution from aerosol scattering that
will shorter the expected light path. On the other hand, in the presence of a brighter surface, enhanced multiple scattering
between the surface and the aerosols may lead to a longer light path. With a RT model, this coupling effect can be explicitly
characterized. In general, in the presence of aerosols, XCO$_2$ (XCH$_4$) will be overestimated if scattering is not accounted for.
This is because underestimation of O$_2$ SCD is larger than that of CO$_2$ (CH$_4$) SCD, due to higher AOD at the lower wavelength,
assuming surface reflectance is relatively unchanged between the two bands. In fact, the reflectance ratio between the 1.6 μm
CO$_2$ band and the 1.27 μm O$_2$ band is about 0.5–0.8, depending on the surface target, according to GFIT3 estimates; the
corresponding AOD ratio is about 0.8, according to MERRA reanalysis data. If the reflectance ratio is close to 1 (no spectral
dependence), the XCO$_2$ (XCH$_4$) bias will be primarily determined by the AOD ratio. However, if the reflectance ratio is small
(strong spectral dependence), the surface is much darker in the CO$_2$ band than in the O$_2$ band. In extreme circumstances, it is
possible for the surface darkening effect to be more dominant than the reduction in SCD underestimation due to lower AOD.
For example, the West Pasadena location is special in that it is close to a park, which has different land use types compared to
the other surface targets. This target has a much lower reflectance ratio than other locations (**Figure A5**), which may explain
the underestimation by CLARS-GFIT compared to GFIT3 for this location, as seen in **Figure 14(c)** and **Figure A4**.



### 7. Conclusions

In this study, we developed GFIT3, a full physics algorithm to retrieve trace gases in the presence of aerosols, and demonstrated its performance by retrieving $XCO_2$ and $XCH_4$ from CLARS-FTS measurements. This algorithm simultaneously retrieves fine and coarse mode aerosol properties including AOD and ALH. Inverse experiments based on synthetic spectra indicate that the uncertainty in CLARS-FTS retrievals of $XCO_2$ and $XCH_4$ due to uncertainty in the RTM, aerosol scattering, and atmospheric profile, which constitute the three most important sources of error, is less than 1% (or less than ~4 ppm for $XCO_2$ and ~20 ppb

for $XCH_4$). The retrieval uncertainty for real CLARS-FTS observations is partly due to imperfect characterization of aerosol properties. Nonetheless, we find that the retrieved AOD has good agreement with AERONET measurements. Future research will focus on developing a "divide and conquer" algorithm for retrieving aerosol properties and GHGs in order to further improve the accuracy of GHG retrievals. The basic idea is to use a two-step procedure. First, $O_2$ absorption lines will be used to constrain the AOD and ALH based on a spectral sorting technique (**Zeng et al., 2019**). These values will then provide

constraints for AOD and ALH (with uncertainty estimates) for the retrieval of GHGs.



**Author responsibilities**: ZZ designed this study, conducted the experiments, analysed the results, and wrote the paper; VN developed the O-PCA code and provided guidance on radiative transfer modelling; FX provided guidance on optimal estimation; SC assisted with the radiative transfer calculations; FG analysed the aerosol observations and made comparisons; 450  TP collected the CLARS data; KS contributed to spectroscopy analysis; GT provided guidance on using GFIT; YY and SS supervised this study. All authors proofread and commented the paper.

**Acknowledgement:** A portion of this research was carried out at the Jet Propulsion Laboratory, California Institute of Technology, under a contract with the National Aeronautics and Space Administration (80NM0018D0004). The MiniMPL 455  data are available from the NASA Megacity project data portal: https://megacities.jpl.nasa.gov/portal/. AERONET data for the Caltech site are available from https://aeronet.gsfc.nasa. gov/new_web/photo_db_v3/CalTech. html. We also thank Jochen Stutz from UCLA and his staff for their effort in establishing and maintaining the AERONET Caltech site. CLARS-FTS data are available from the authors upon request, and part of the data is available from the NASA Megacities Project at https://megacities.jpl.nasa.gov.




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

**Figure 1. (a) Schematic figure of the CLARS observatory. The lines depict incident and reflected sunlight from an example surface reflection target. The LABS and SVO observation modes are illustrated. For the LABS mode, examples of contributions from single scattering (dotted-red) and multiple scattering (dotted-black) are also illustrated; (b) Examples of observed high resolution (0.06 cm⁻¹) spectra for the O₂ ¹Δ absorption band centered at 1.27μm (7885 cm⁻¹), the weak CO₂ absorption band at 6220 cm⁻¹, the CH₄ absorption band at 6076 cm⁻¹, and the strong CO₂ absorption band at 4852 cm⁻¹. These measurements were made on 28 September, 2013 over the Santa Anita Racetrack surface reflection point at local noon; (c) The 33 surface reflection points across the Los Angeles basin. The background image is adopted from © Google Earth.**





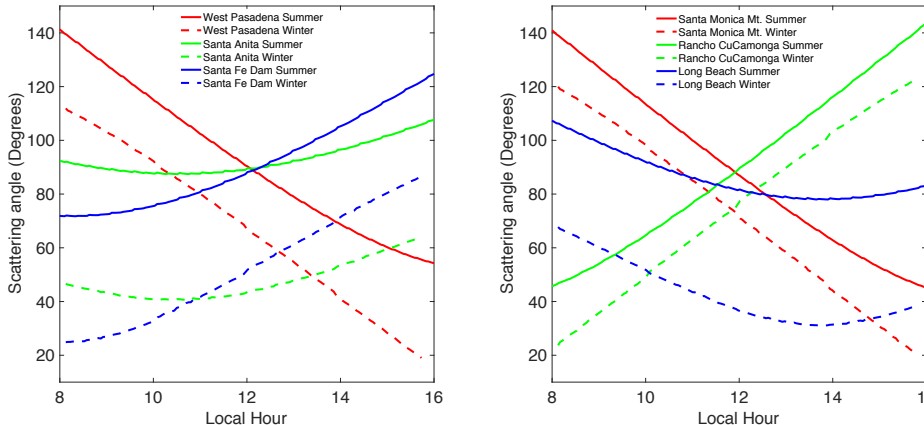

**Figure 2. Diurnal change of aerosol scattering angle for six selected surface reflection points, separated into two groups. Group 1 includes points #1 Santa Anita Racetrack, #2 West Pasadena, and #3 Santa Fe Dam that are close to CLARS-FTS; group 2 includes #15 Santa Monica Mt., #17 Rancho Cucamonga, and #19 Long Beach that are further away. Hourly scenarios from June 20 and December 22 2013 are used to represent summer and winter solar geometries, respectively.**





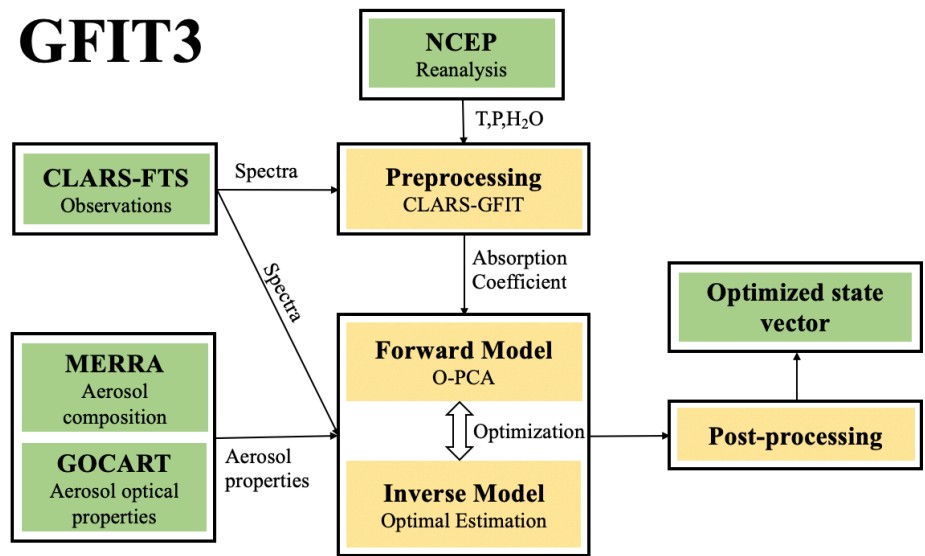

**Figure 3. Workflow of GFIT3 for retrieving XCO2 and XCH4 from CLARS-FTS observations. There are four major components: (1) a pre-processing step to identify soundings free of clouds and heavy aerosol loading; (2) a forward RTM to generate synthetic spectra in order to simulate observed CLARS-FTS measurements; (3) an inverse model based on optimal estimation to update the surface and atmosphere states to minimize the difference between model and observation; and (4) a post-processing screening step to filter out bad retrievals.**




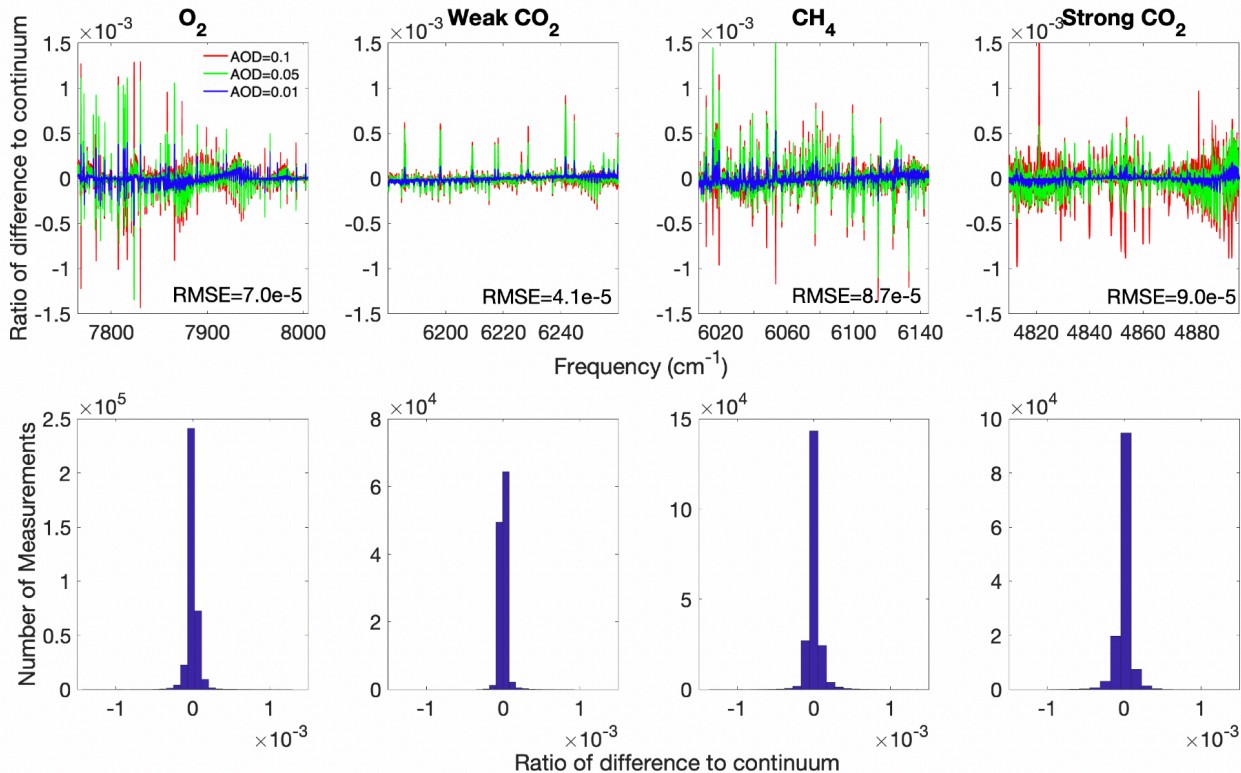

**Figure 4. Ratio of the difference (relative to the continuum value) between simulated radiances (using O-PCA) and high-accuracy computations (using LIDORT with 32 streams). These calculations are based on the 240 scenarios, with different observation geometries and atmospheric profiles, described in the inverse experiments in Section 4. Four EOFs are used for the O-PCA computations. Three different aerosol scenarios are considered, with AOD of 0.01, 0.05, and 0.1, respectively, in the 1.27 μm $O_2$ $^1\Delta$ band. The overall RMSEs are also indicated.**





(a)                                                      (b)

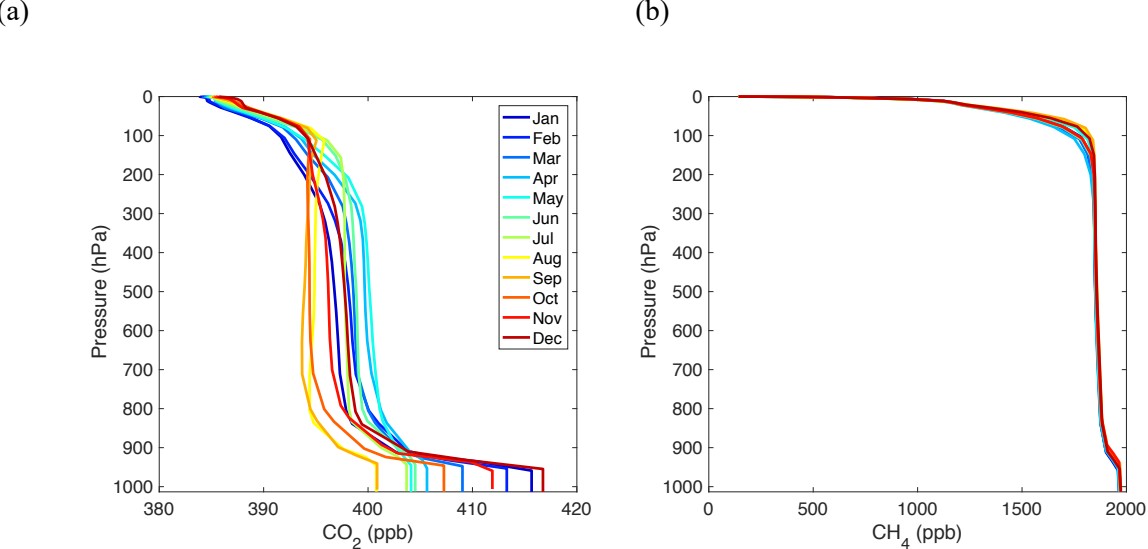


**Figure 5. (a) CO₂ vertical profiles are extracted from the CarbonTracker model over Los Angeles with 3-hourly temporal resolution. Monthly averaged profiles are used as *a priori* in GFIT3. (b) Monthly averaged CH₄ vertical profiles are adopted from CLARS-GFIT. A constant PBL enhancement of 91 ppb, as estimated by Verhulst et al. (2017; Table 5) using the NASA Megacity network, is added. The hourly variability of CH₄ in the PBL is assumed to be the same as that of CO₂ since they are co-emitted and follow a similar atmospheric mixing process.**





(a)                                                   (b)

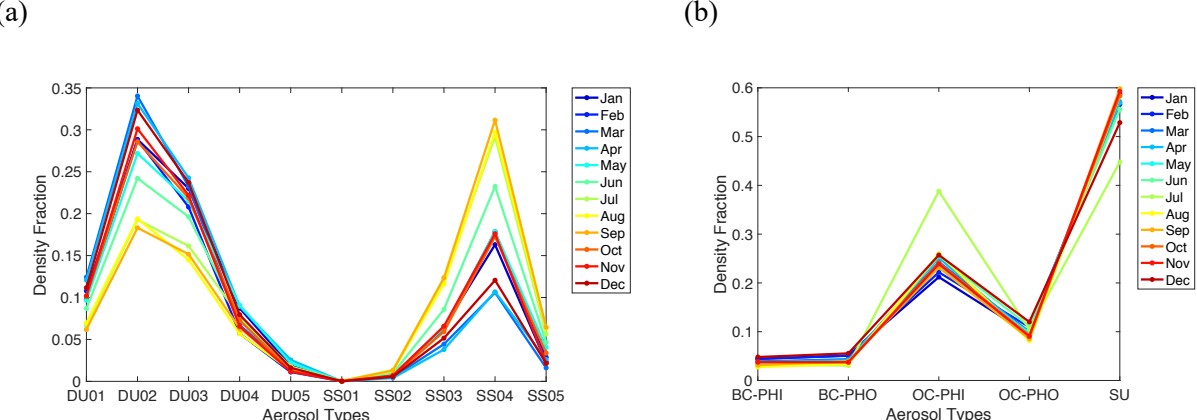

**Figure 6. Aerosol composition from Modern-Era Retrospective Analysis for Research and Applications (MERRA) reanalysis data for LA (local times 7h,10h,13h,16 h). (a) Monthly averaged density fraction of aerosols for dust and sea salt. The dry size bins for dust (DU01 to DU05) correspond to the radius limits (in microns) 0.1–1, 1–1.8, 1.8–3, 3–6, and 6–10, respectively. Similarly, for sea salt (SS01 to SS05), the corresponding values are 0.03–0.1, 0.1–0.5, 0.5–1.5, 1.5–5, and 5–10, respectively; (b) Monthly averaged density fraction for hydrophilic black carbon (BC_PHI), hydrophobic black carbon (BC_PHO), hydrophilic organic carbon (BC_PHI), hydrophobic organic carbon (BC_PHO), and sulfate (SU). MERRA data below the CLARS-FTS elevation (1.67 km) are used.**





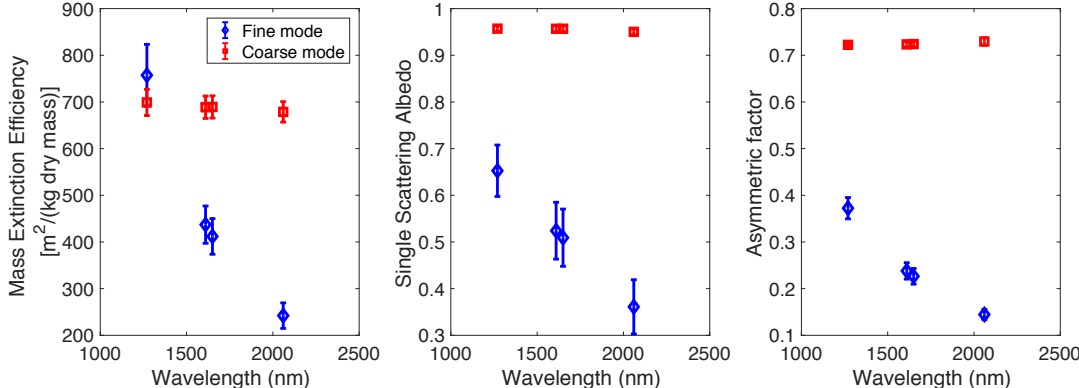

**Figure 7. Wavelength dependence of aerosol optical properties (averaged over a year) in the 1.27 um $O_2$ $^1\Delta$ absorption band, 1.61 μm weak $CO_2$ absorption band, 1.65 μm $CH_4$ absorption band, and 2.06 μm strong $CO_2$ absorption band from the Georgia Institute of Technology–Goddard Global Ozone Chemistry Aerosol Radiation and Transport**
**(GOCART) model. (left) mass extinction efficiency; (center) single scattering albedo; and (right) asymmetric factor for fine (blue) and coarse (red) modes. These aerosol optical properties are density weighted on a monthly basis for day time only (local times 7h, 10h, 13h, 16h). For aerosols that are hygroscopic (size dependent upon relative humidity), monthly average humidity is used.**





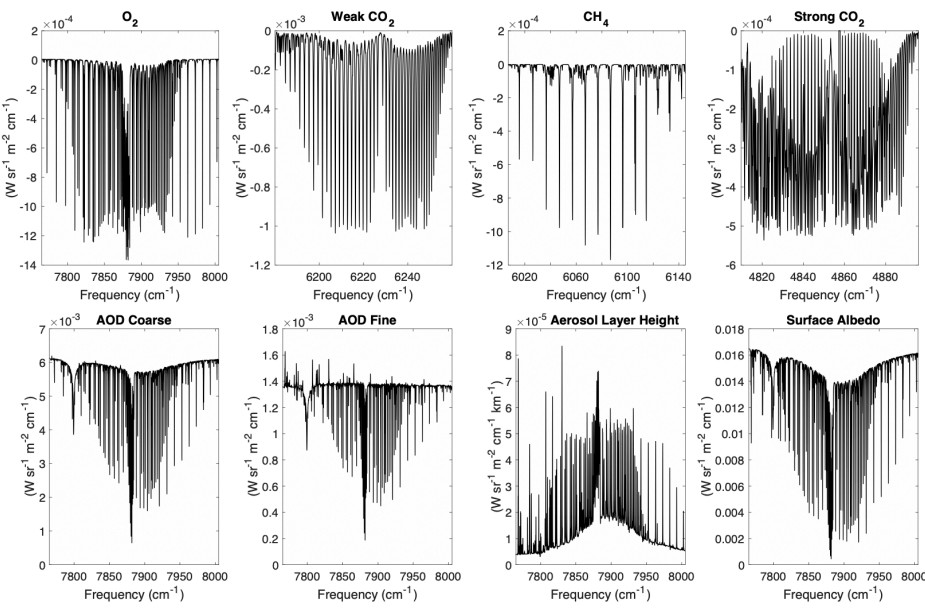


**Figure 8. Sample Jacobian values from O-PCA for representative state vector elements in the GFIT3 retrieval algorithm. This Jacobian matrix is based on observations over the Santa Anita surface reflection point on September 28, 2013, with a solar zenith angle (SZA) of 36 degrees. The y-axis labels indicate the units of the Jacobian values.**


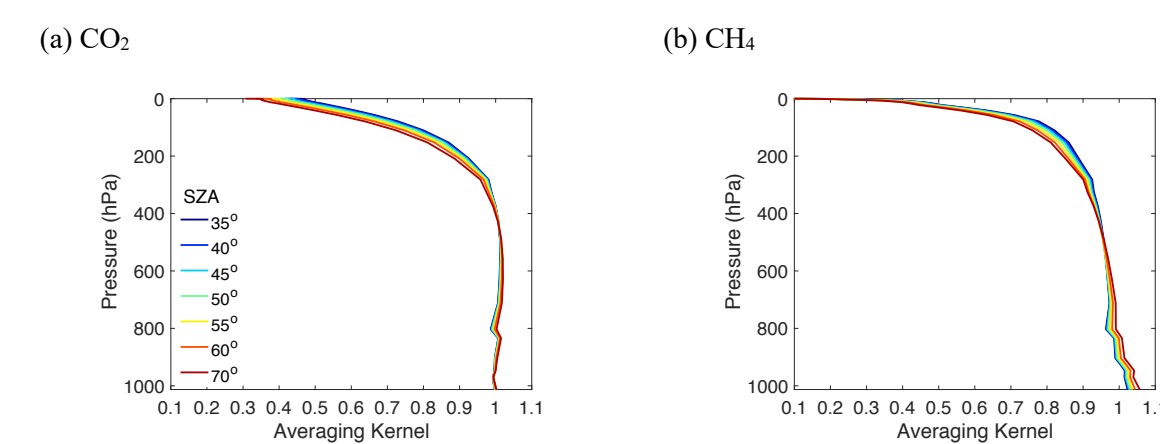

**Figure 9. Examples of column averaging kernels for (a) CO₂ and (b) CH₄ with different SZA. These are from observations of the Santa Anita surface target on September 28, 2013.**


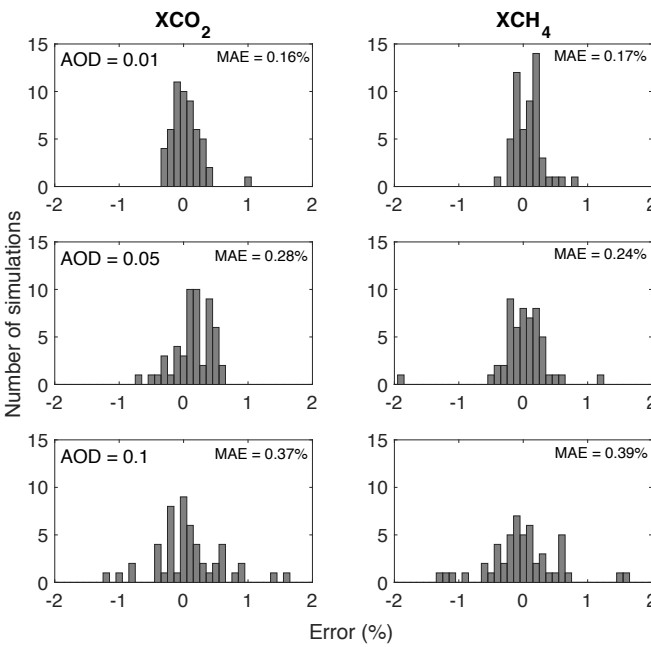

**Figure 10. Results for Test 1. Errors in retrieved XCO₂ and XCH₄ are quantified for simulations with three different values of AOD (0.01, 0.05, and 0.1). The errors arise mainly due to the bias caused by the O-PCA approximation compared to the exact atmospheric radiative transfer process. MAE represents the mean absolute error.**






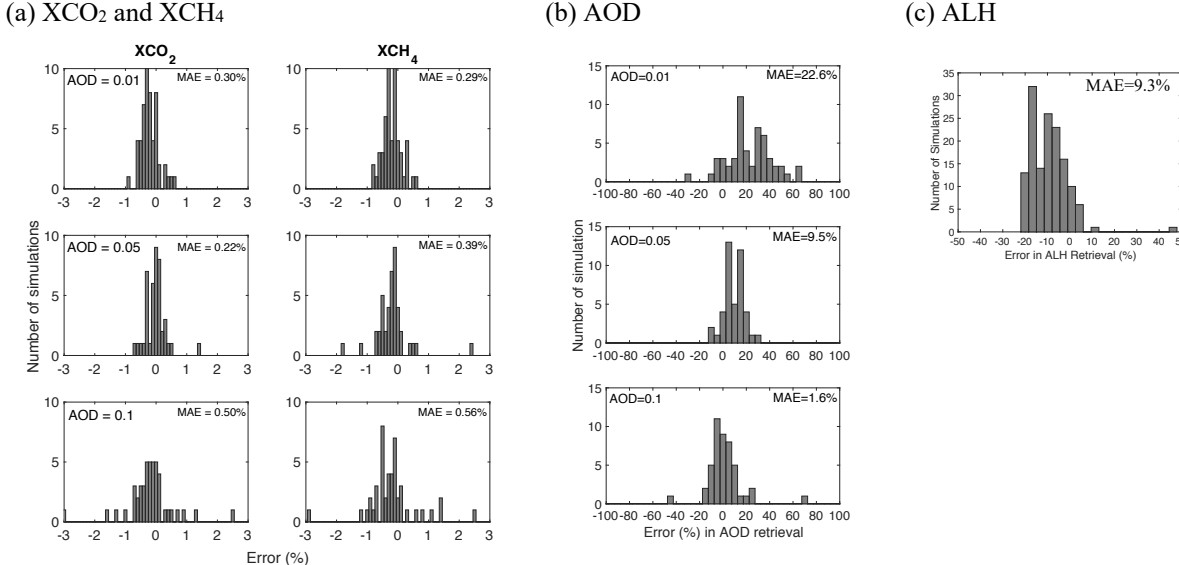

**Figure 11. Results for Test 2. Errors in retrieved (a) XCO₂ and XCH₄ for three different values of AOD (0.01, 0.05, and 0.1); (b) AOD for the same scenarios as in (a); (c) ALH. The errors have contributions from biases due to the O-PCA**
**RTM and due to imperfect knowledge of aerosol optical properties and vertical distribution of atmospheric trace gases.**



(a) Biases due to biases in SSA and *g* of coarse mode aerosol

(b) Biases due to biases in SSA and *g* of fine mode aerosol

(c) Bias due to biases in atmospheric profile

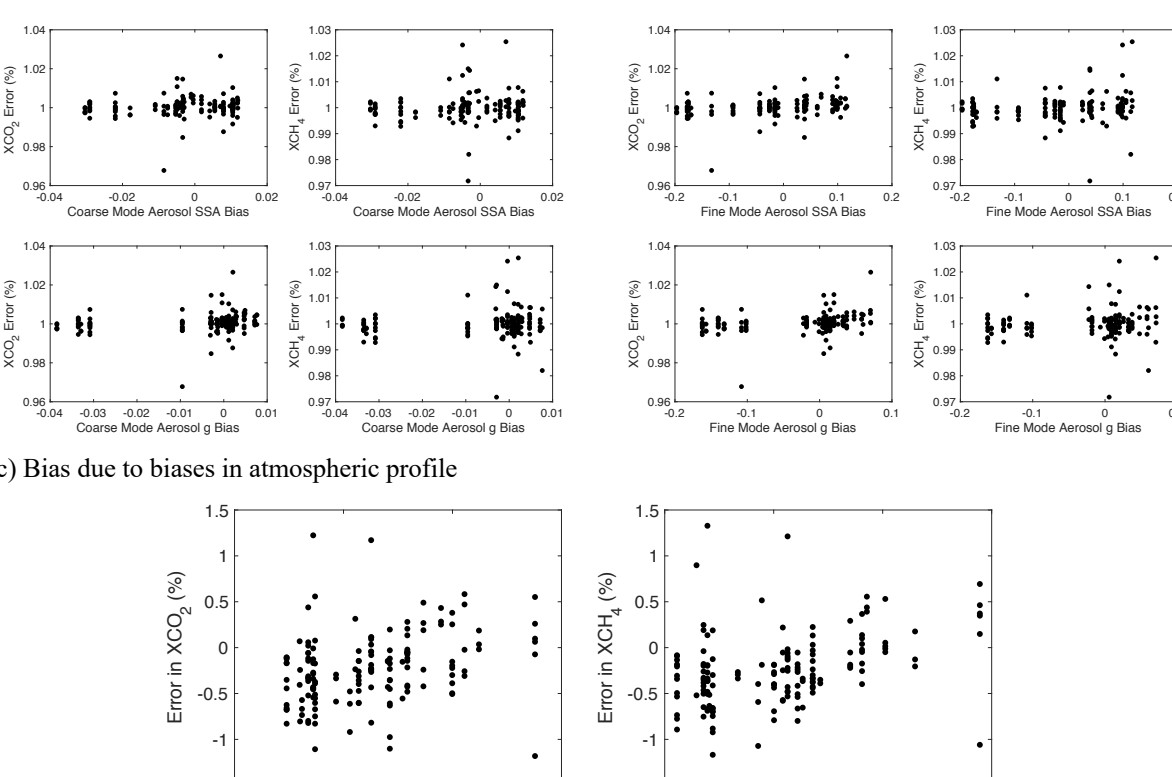

**Figure 12. Results for Test 3 (panels a and b), and Test 4 (panel c). (a) and (b) show XCO₂ and XCH₄ biases as a function of biases in SSA and *g* (asymmetric factor) of coarse and fine mode aerosol, respectively. (c) shows the same as a function of biases in PBL CO₂ and CH₄ enhancement.**







**Figure 13. (a) Upper left: median fitting residual (black) and ±1-σ range (grey) for the $O_2$ band; Lower left: sample measured spectrum; Right: histogram of fitting residuals; (b), (c), and (d) are the same as (a) but for the weak $CO_2$ band, the $CH_4$ band, and the strong $CO_2$ band, respectively.**







**Figure 14. Comparison of (left) XCO₂ and (right) XCH₄ retrievals from GFIT3 and CLARS-GFIT for the (a) Santa Anita; (b) Santa Fe; and (c) West Pasadena surface reflection targets. The data points are color-coded by the retrieved AOD. The insets show scatter plots between retrieved AOD and the difference in XCO₂ or XCH₄ between GFIT3 and CLARS-GFIT.**



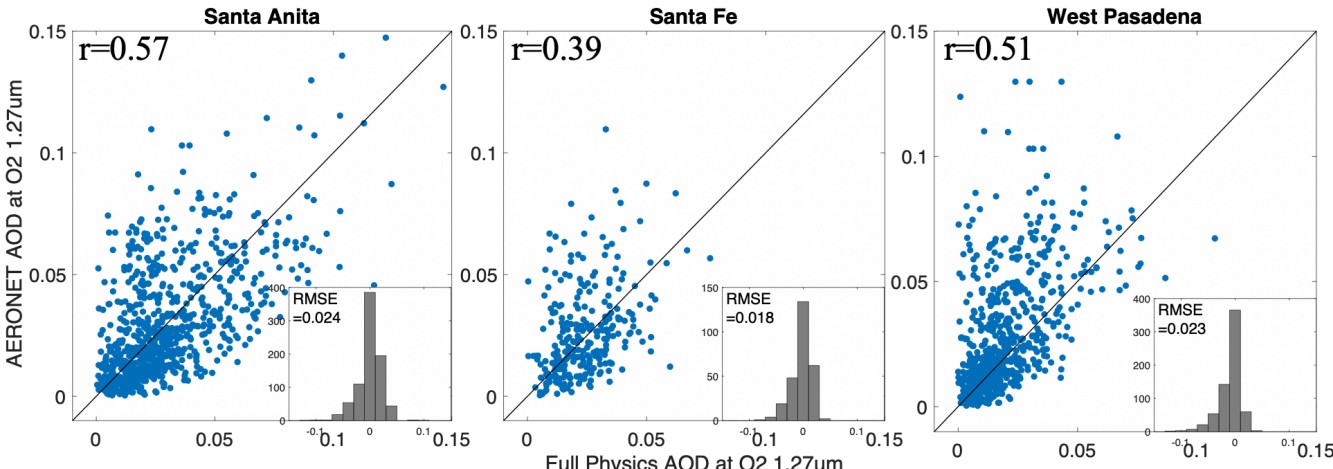

**Figure 15. AOD comparison between measurements from the Caltech AERONET site and GFIT3 retrievals. The AERONET AOD at 1.27 μm is extrapolated from actual AERONET observations using the Angstrom exponent law. Histograms of the difference between AERONET and GFIT3 retrievals are also included.**


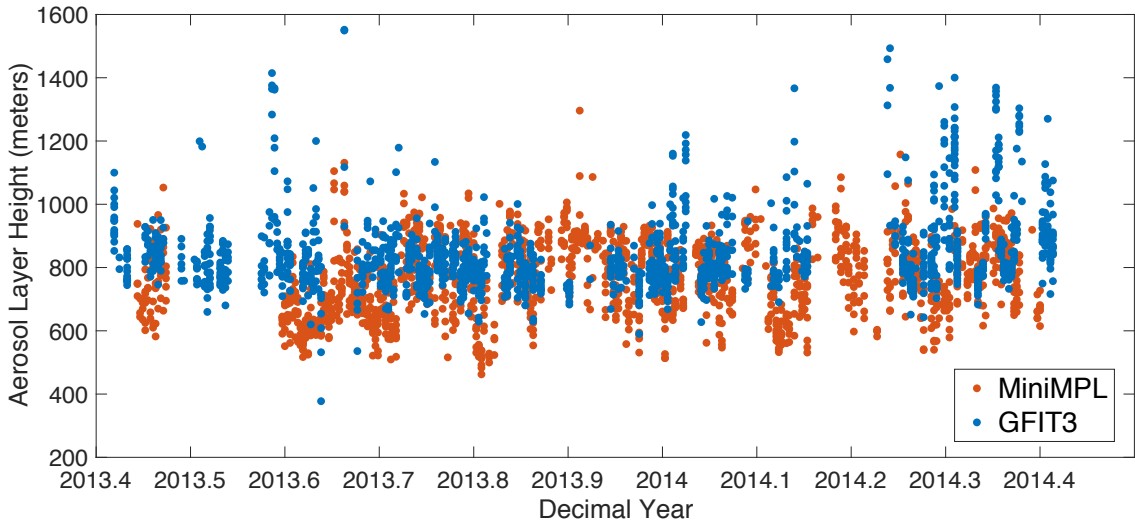

**Figure 16. Comparison of effective ALH from the MiniMPL lidar instrument on the Caltech campus and GFIT3 retrievals for the Santa Anita, Santa Fe, and West Pasadena surface targets.**




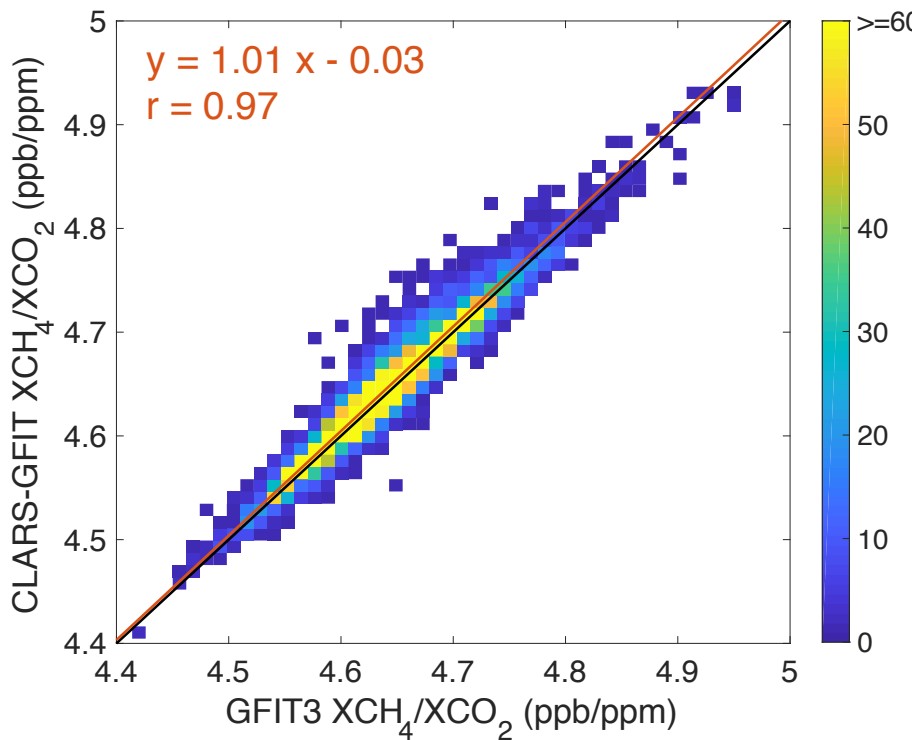

**Figure 17. Scatter plot of the XCH₄/XCO₂ ratio from GFIT3 and CLARS-GFIT. The 1:1 line is shown in black. The red line denotes the best fit using type II linear regression to fit the data. The equation for the regression fit is also shown.**






## Appendix Figures

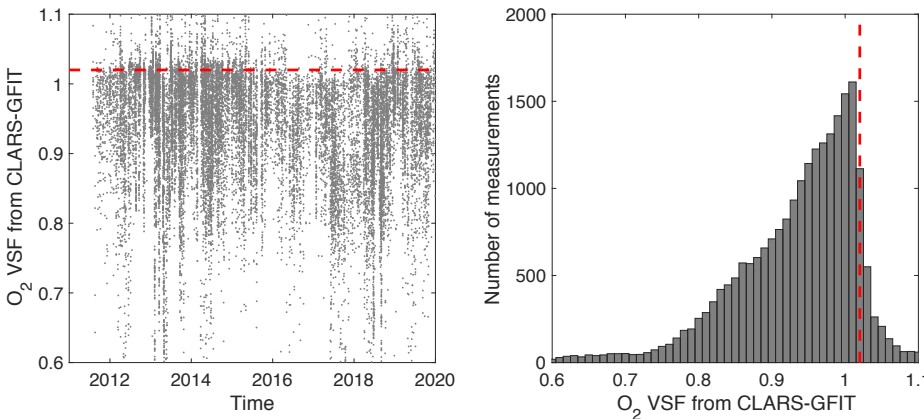

**Figure A1. (left) Time series of O₂ VMR Scale Factor (VSF) and (right) histogram of VSF. The VSF value (indicated by the dashed red line) of ~1.02 represents situations when the atmosphere is clear. See Section 3.1.1 for details.**

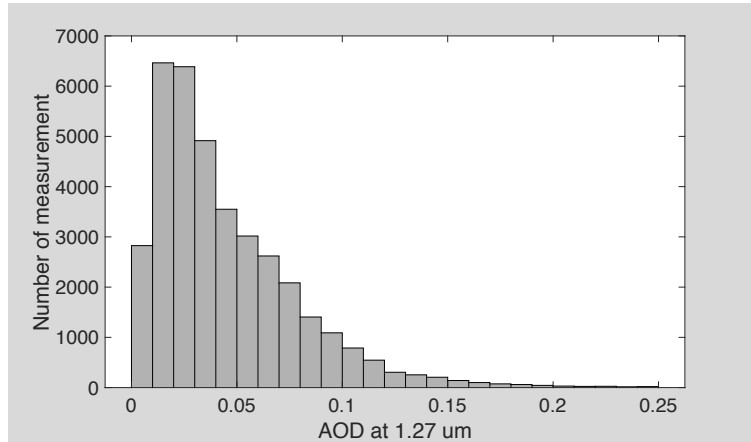


**Figure A2. AOD in the 1.27 μm O₂ absorption band estimated from AERONET observations (2010–2017).**





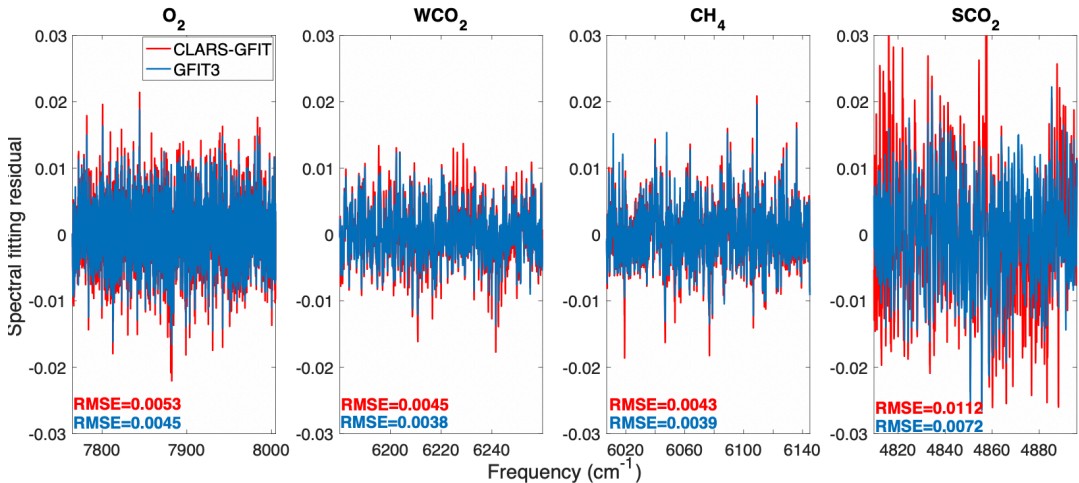


**Figure A3. Example of spectral fitting residuals from the CLARS-GFIT (red; ignoring aerosol scattering) and GFIT3 (blue; accounting for aerosol scattering) algorithms for the O₂, WCO₂, CH₄, and SCO₂ spectral windows. The spectral fitting RMSEs are also indicated. This example is for an observation over the West Pasadena surface target on Sept. 28, 2013 with a solar zenith angle of 65°.**






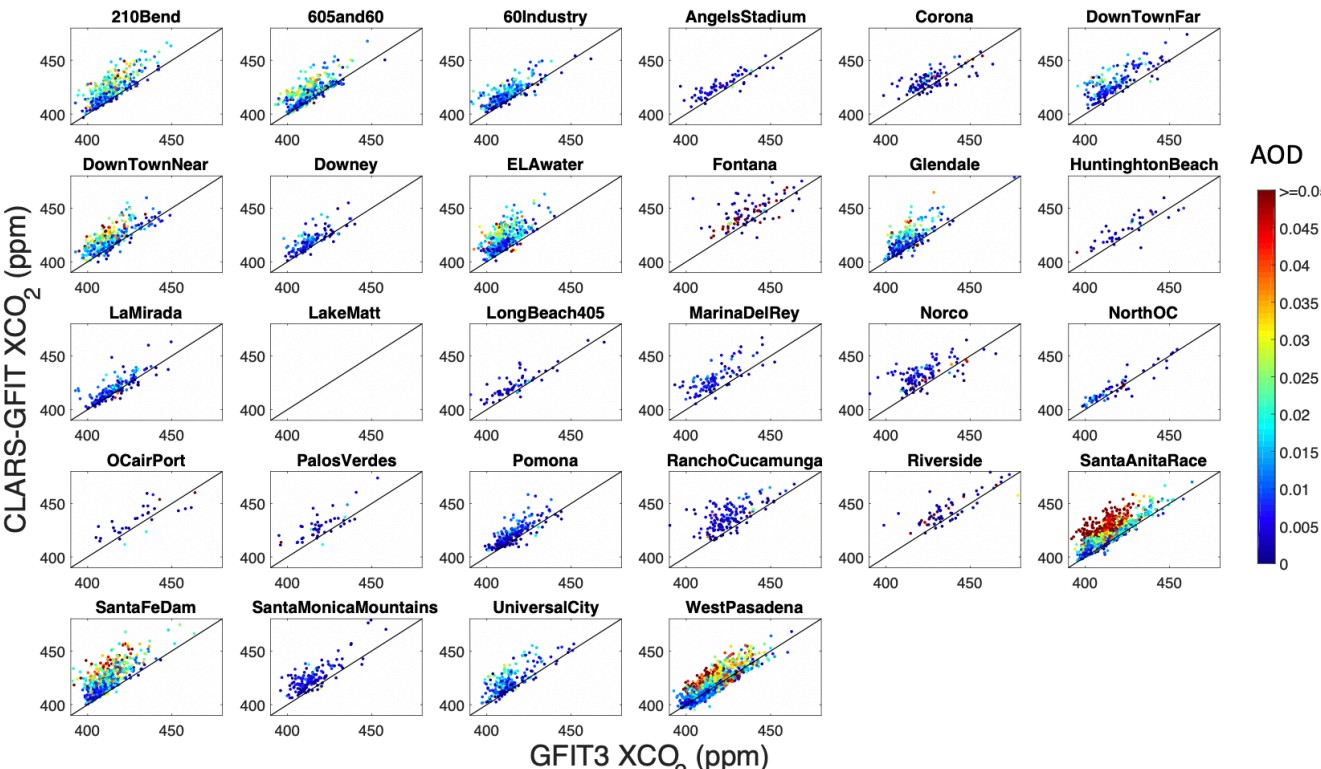

**Figure A4. Comparison of XCO₂ retrievals from GFIT3 and CLARS-GFIT for all surface reflection targets. The data points are color-coded by the retrieved AOD.**





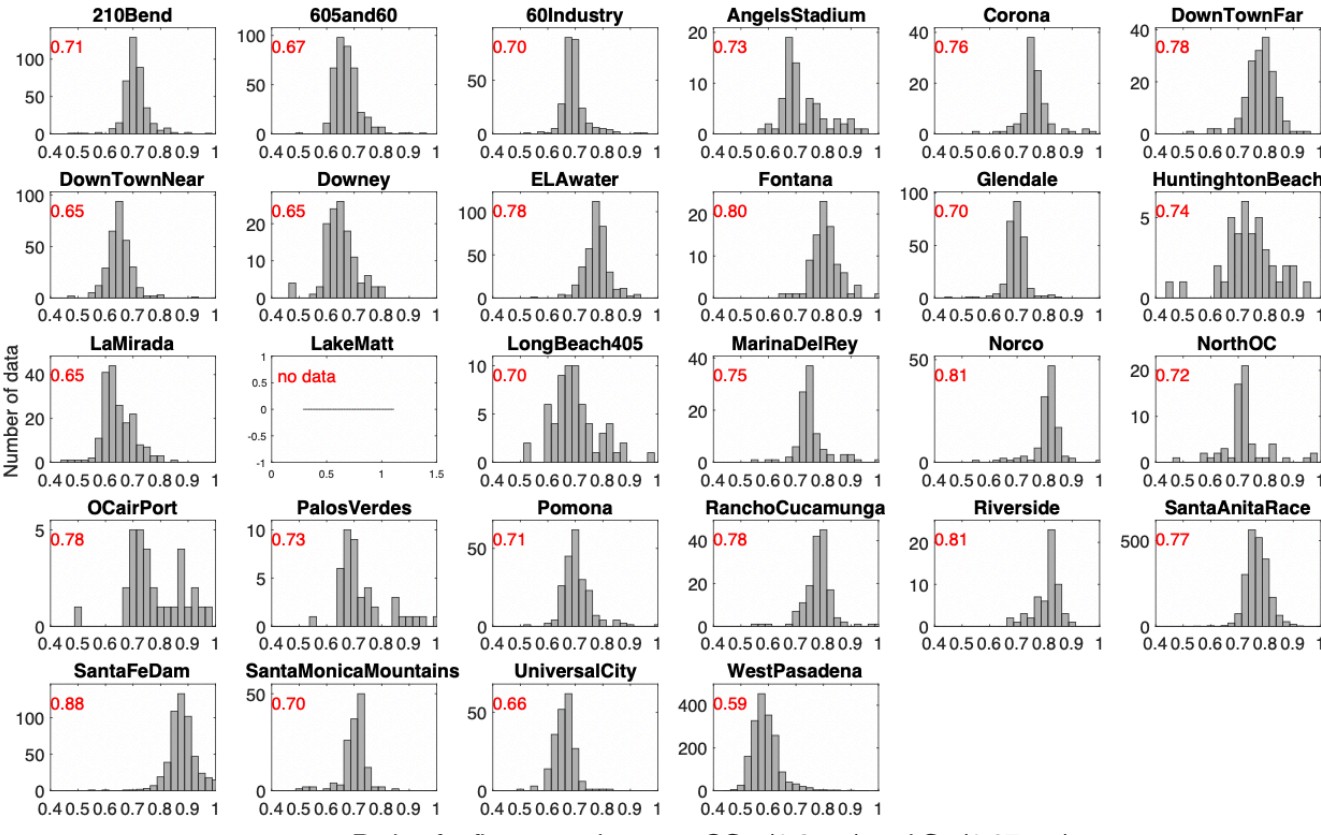


**Figure A5. Histogram of the ratio of reflectance between the WCO₂ and O₂ bands for all the surface targets. The reflectance values are obtained from GFIT3 retrievals. The number in red is the average ratio for the surface target.**
