# Peer review of "GFIT3: A full physics retrieval algorithm for remote sensing of greenhouse gases in the presence of aerosols"

_Atmospheric Measurement Techniques, 2021_

## Author Comment (AC1)

**RC1: 'Comment on amt-2021-84', Anonymous Referee #1, 07 May 2021**
This is an informative and well-written paper on a subject which is important to understanding remote sensing of greenhouse gases, and should certainly be published after some revision.

Thank you for your positive comments and constructive suggestions. Please see our item-by-item responses, and associated changes (highlighted in yellow) in the revised manuscript, to your comments and suggestions below.

My suggestions and questions follow: First I will discuss two important points that I believe need explanation and a bit more consideration by the authors. Pages 8 and 9 of the manuscript describe the state vector and its relation to the underlying forward model. So for example a $CO_2$ a priori profile is assumed for the forward model and a single scale factor for the profile is retrieved in the state vector. For aerosol, values of AOD are retrieved for coarse and fine aerosols, and a single layer height value is retrieved. It is not at all clear how those 3 scalar quantities are related to the underlying quantities of albedo, phase function, and altitude distribution of the 5 aerosol types discussed in lines 205-224. Please explain. Are the values of $CO_2$ and $CH_4$ affected by the details of this assumed relationship?

All elements (GHG profile scale factors, AODs, and ALH) in the state vector are coupled. On the other hand, the single scattering albedos (SSAs) and phase functions of the aerosol coarse and fine modes are prescribed and not retrieved. The effective SSA for the coarse mode is calculated as the mean of the SSA values (from the GOCART model) of sea salt and dust, weighted by their simulated AODs from MERRAero. The same methodology is applied to fine-mode aerosols except using black carbon, organic carbon and sulfate. The effective phase functions can be calculated in a similar manner, except that the weighting is done by the scattering AOD.

We do not consider the geometric thickness of the aerosol layer since it has much smaller impact on the observed radiance compared to the total AOD (Zeng et al., 2019). Practically, in the forward model, the aerosols are placed in two adjacent layers. The fractions of AODs in each layer are adjusted (with total AOD conserved) to change the effective ALH. Since both fine- and coarse-mode aerosols are relatively well mixed in the atmosphere, we assume that they have the same effective ALH.

Errors in the aerosol optical properties will propagate through the forward model and affect the state vector to be optimized. To investigate this problem, a synthetic experiment (Test 3 in Section 4) has been conducted to quantify the impact on the GHG retrievals of imperfect prescription of aerosol optical properties. From this experiment, we conclude that there is no clear correlation between bias in XCO2 or XCH4 retrievals

and that in aerosol optical properties of either coarse- or fine-mode aerosols. This indicates that a combination of fine- and coarse-mode aerosols is able to accurately capture the scattering effects.

We added the above paragraphs in Section 3.2.2 and Section 4.

My second important point concerns uncertainties in the retrieval. Namely, the discussion on p. 10-11 (lines 260-267), and conclusions on p. 18 attribute forward model error entirely to noise and to approximations in the calculations of multiple scattering. But surely, spectroscopic error and uncertainties in aerosol albedo and phase function are far larger. Or if not, that should be stated and justified. What has been done to evaluate these well-known sources of forward model error?

As we stated in the manuscript, $\varepsilon$ in Equation (1) is the error vector containing both the measurement noise and the forward model error. Many previous studies have shown that the measurement noise dominates; however, the forward model error, including multiple components such as errors in spectroscopy and biases in prescribed aerosol optical properties, may not be negligible. These uncertainties propagate through the retrieval algorithm to the retrieved GHGs.

The errors due to imperfect aerosol optical properties (arising from uncertainties in aerosol fraction, size distribution, and microphysical properties) are investigated through a synthetic experiment (Test 3 in Section 4) that quantifies their impact on the GHG retrievals. As in our answer to the previous comment, from our experiment, we conclude that there is no clear correlation between bias in XCO2 or XCH4 retrievals and that in aerosol optical properties of either coarse- or fine-mode aerosols.

The benefit of using the GFIT spectroscopy database is that it has been carefully evaluated based on highly accurate TCCON observations. To further investigate the errors in spectroscopy and their impact on the GHG retrievals, we apply Principal Component Analysis (PCA) analysis on the fitting residuals. This analysis method has been used by the OCO-2/3 operational algorithm to study the effect of imperfect spectroscopy. The three principal components (PCs) with the largest variance are shown in the following figure. The features in these PCs are mostly related to spectroscopic uncertainties. These PCs might be related to line width, instrument effects, and the solar spectrum. For example, PC-3 from the $WCO_2$ band appears to be correlated with absorption features that can be attributed to very small changes in the line width. However, this PC can only explain a few percent of the residual variance. Overall, there are no dominant PCs that can explain more than 10% of the variance in the fitting residual. This is because the fitting residual itself is very close to random and without

large systematic errors. We therefore believe that spectroscopic errors should not be a major issue here.

A summary of the above paragraphs has been added to Section 6.3 in the revised manuscript.

[Figure]

Figure A6. Mean radiance spectrum, and the three leading principal components (PCs), ranked by the variance explained by these PCs, obtained by applying Principal Component Analysis (PCA) on the fitting residuals, for the (a) O2, (b) WCO2, (c) CH4 and (d) SCO2 bands. The variance explained by each PC is also indicated.

A related point concerns the measurement error covariance matrix. The error sources I refer to are strongly correlated between channels. It is nevertheless usual in my my experience to assume a diagonal measurement covariance matrix, for practical reasons. But that assumption should be explicitly recognised as such, and the effect of those errors evaluated post retrieval.

We agree with the reviewer that correlations between channels exist and that a diagonal measurement error covariance matrix is assumed for the sake of simplicity. The correlation between channels may come from several factors, such as phase correction in converting from interferometry to radiance and detector nonlinearity. However, these impacts should be much smaller compared with measurement noise. A comprehensive evaluation of this correlation is beyond the scope of this study. However, the reduced $\chi^2$, which is the $\chi^2$ from equation (2) divided by the total number of measurements and state vector elements, infers the goodness of fit and can be used to evaluate the error covariance matrix. Theoretically, if the error covariance matrix is properly implemented in the retrieval algorithm, the reduced $\chi^2$ should be close to 1 after convergence, which means that the fitting residuals are consistent with the detector noise estimates. The histogram of reduced $\chi^2$ from all converged retrievals (see below figure) indicates that most of the retrievals have a $\chi^2$ close to 1, with 83% having $\chi^2$ less than 1.5. This indicates that the error covariance matrix used in the retrieval algorithm, which assumes that measurement noise is uncorrelated between spectral channels, is realistic. We also admit that inaccuracies in the spectroscopic input data and improperly modeled instrument effects may contribute to the small deviation of $\chi^2$ from unity. Related statements have been added to Section 6.3 in the revised manuscript.

[Figure]

**Figure A7**. Histogram of reduced $\chi^2$ from all converged retrievals in this study.

Aside from the two major points above, I have a few minor ones:

p. 2-3, lines 62-68: There is a very recent paper on GFIT2 which the authors will know about (it shares 1 co-author with this paper):

Roche, S., Strong, K., Wunch, D., Mendonca, J., Sweeney, C., Baier, B., Biraud, S. C., Laughner, J. L., Toon, G. C., and Connor, B. J.: Retrieval of atmospheric $CO_2$ vertical profiles from ground-based near-infrared spectra, Atmos. Meas. Tech., 14, 3087–3118, https://doi.org/10.5194/amt-14-3087-2021, 2021.

This is very relevant. It has been added to the references.

p. 14, line 360, and Fig A4: The three reflecting points most used are also the closest to the instrument. I presume that explains why Fig A4 shows the AODs for other sites have mostly low values, in comparison to the AOD values for the 3 dominant sites. A sentence or two would be interesting to confirm and interpret that.

We add the following sentences in Section 5.3.

**In comparison to the three sites close to the CLARS location (Santa Anita, Santa Fe, and West Pasadena), for sites that are further away, valid retrievals that pass the filters have lower AOD values. This is because of their longer slant paths in the PBL, leading to a larger scattering effect even under the same vertical aerosol loading.**

p. 15, line 397: CLARS-FP?

Corrected. It should be GFIT3.

p. 17: Section 6.2 is confusing. The discussion jumps back and forth from one assumption to another about reflectance ratio, and loses this reader.

We have substantially rephrased this paragraph. Please see Section 6.2 in the revised manuscript.

---

## Author Comment (AC3)

**RC2: ['Comment on amt-2021-84'](), Anonymous Referee #2, 18 May 2021**

This paper presents progress towards implementing an aerosol retrieval to the existing CLARS-GFIT algorithm. The CLARS-FTS instrument is a peculiar ground-based FTS as it collects solar spectra from reflected sunlight rather than observing the sun directly. Because of the much lower signal with reflected sunlight, the effect of aerosols must be included in the forward model. The paper presents experiments with the new algorithm to estimate the effect on retrieved GHGs of simultaneously retrieving the aerosol optical depth and aerosol layer height.

Thank you for your positive comments and constructive suggestions. Please see our item-by-item responses, and associated changes in the revised manuscript, to your comments and suggestions below.

This article should be published after addressing the following comments and questions:

**General comments**:

GFIT3 may be a confusing name. GFIT2 enables profile retrievals and has a separate implementation of the inverse method. I would assume a "GFIT3" would add something to GFIT2.

GFIT3 is significantly different from GFIT2 in the following two ways. **First**, GFIT3 is primarily used for top-down observations that measure surface reflected solar radiation. It simulates the incident sunlight from the sun to the surface, and then reflected from the surface to the instrument (CLARS-FTS in this study). It can also be used for existing space-borne instruments like GOSAT and OCO-2/3. **Second**, because of the strong aerosol scattering effect, an aerosol model (based on MERRAero) and a fast radiative transfer model (O-PCA) that enables the simulation of multiple scattering due to aerosols are incorporated in the simulation. On the other hand, GFIT and GFIT2 are designed for directly transmitted solar spectra. In this case, the scattering effect is negligible, so there is no need to account for it. The following sentences are added in the revised manuscript:

**"GFIT3 is designed specifically for the purpose of retrieving GHGs in polluted atmospheres from reflected solar radiation observations. It includes an aerosol model and a fast radiative transfer model to simulate the aerosol scattering contributions."**

I feel section 5.3 should come before Section 5.2 as it should be shown that the retrieved AOD compares well to independent measurements (and is an improvement from the a priori AOD) before comparing CLARS-GFIT and GFIT3 GHG retrievals.

This is a good suggestion. We swapped these two sub-sections in the revised manuscript.

Is CLARS-FTS able to target the Pasadena TCCON site located at CalTech? Can it be used to validate its GHG retrievals? This could be discussed in the text.

Unfortunately, due to mountain ridges in the line of sight, CLARS-FTS cannot directly target the TCCON site at Caltech. On the other hand, TCCON uses directly transmitted solar spectra to measure GHG columns, which have different observing geometries from CLARS observations, as shown in the figure below (red for CLARS and black for TCCON). Therefore, a direct comparison is not feasible. The interpretation of the comparison would need to consider the spatial heterogeneity of GHG distributions between the incident and reflected solar paths in the boundary layer. Therefore, to make such a comparison, we would need to reconcile the difference in observing geometries, which is beyond the scope of this study.

We added the following sentences in the revised manuscript:

**"Unfortunately, direct comparison of XGHGs with existing TCCON data at Caltech is not feasible. On one hand, CLARS-FTS cannot directly target the TCCON site at Caltech due to mountain ridges that block the line of sight. On the other, TCCON uses directly transmitted solar spectra to measure GHG columns, which have different observing geometries from CLARS observations; the spatial heterogeneity of GHG distributions between the incident and reflected solar paths in the boundary layer make the results difficult to compare."**

[Figure]

Figure. Comparison of observing geometries of CLARS, TCCON, and satellites.

How much more time does it take to run a retrieval with LIDORT 32 streams compared to O-PCA? Figs. 10 and 11 seem to show that the effect on $XCO_2$ and $XCH_4$ MAE of using the less accurate model is comparable to switching from 3-hourly to monthly a priori GHG profiles and aerosol composition.

The computation time for a single retrieval using O-PCA is about 30 times less than that using LIDORT, which is a significantly improvement.

If we focus on the high aerosol loading scenario (AOD=0.1), we can see that the error (MAE) in Figure 11 (0.50% and 0.56% for $CO_2$ and $CH_4$, respectively) is larger than that in Figure 10 (0.37% and 0.39%). This indeed indicates that imperfect aerosol optical properties lead to high uncertainty in the GHG retrievals.

**Specific comments:**

Table 1: add units for aerosol layer height.

Added in the revised manuscript.

Figure 3: is the post-processing step really done before the final iteration of the optimal estimation method that yields the a posteriori state vector? Is the retrieved state changed during post-processing and becomes the "optimized state vector"?

We switch the order of "Post-processing" and "optimized state vector" in the revised manuscript.

Figures 11(c): it looks like the ALH panel has more simulations than the other panels and more than the "60 different observation scenarios" mentioned on Line 337. Is it the sum from simulations with all AODs? Clarify in the text or in the caption.

Yes, it is the sum of all cases. We added the description in the figure caption.

**"(c) ALH from all tests with different AODs."**

The "asymmetric factor" is only included in captions of Figs. 7 and 12, mention in the text what it is and why it is important.

We added the explanation in the text:

**"To illustrate changes in phase function, the asymmetric factor (that quantifies the extent of forward scattering) is used. An asymmetric factor of 0 represents**

**isotropic scattering; the value increases to 1.0 as the phase function peak sharpens in the forward direction."**

Line 46: spell out AERONET.

Added. Thanks.

Line 63-64: specifying that GFIT2 uses optimal estimation makes it sound like GFIT does not, both GFIT and GFIT2 use the optimal estimation method.

We rephrased the descriptions in the text:

**While GFIT scales profiles based on optimal estimation, GFIT2 uses Bayesian optimal estimation theory (Rodgers, 2000) as the inverse method to retrieve GHGs at different altitudes.**

Line 85-86: I feel there should be a sentence to indicate if airglow would also have an impact on CLARS-FTS measurements.

CLARS-FTS looks downwards toward the basin, so it is not affected by airglow, which is an upper atmospheric phenomenon. We added this statement in the revised manuscript.

**"Since CLARS-FTS looks downwards toward the basin, the measured spectra are not affected by airglow, which occurs in the upper atmosphere."**

Line 139: given the upcoming release of a new GGG version with major updates to the spectroscopic linelist, it should be mentioned here that you have been using the GGG2014 linelist (if this is what you used).

We added the version information (GGG2014) in the revised manuscript, and indicate that we will update GFIT3 to the latest GGG version (GGG2020) in the near future.

Line 143: is (4) referring to the RMS of fit residuals? 1-sigma above the mean of what? The mean of the RMS of residuals from multiple spectra? The mean of the residuals of the current spectrum?

Added. It is the mean of all fitting residuals.

**"(4) spectral fit error larger than one sigma above the mean of all the spectral fitting residuals."**

Line 149: Washenfelder et al. (2006) is missing from the reference list.

Added in reference:

Washenfelder, R. A., Toon, G. C., Blavier, J. F., Yang, Z., Allen, N. T., Wennberg, P. O., Vay, S. A., Matross, D. M. and Daube, B. C.: Carbon dioxide column abundances at the Wisconsin Tall Tower site. J. Geophys. Res. Atmos., 111(D22). https://doi.org/10.1029/2006JD007154, 2006.

Line 223-224: the a priori ALH is derived from an aerosol profiling lidar, but the a priori value in Table 1 is 0.7 (km?), does the a priori change based on the MiniMPL observations or were these observations used to determine a fixed 0.7 would be used as a priori value?

We used the averaged value (0.7 km) as the a fixed a priori for all retrievals, because the coverage of the MiniMPL is not long enough to derive the climatological change with time in LA. We added the following sentence in the revised manuscript:

**"For the retrievals, the *a priori* ALH is set to 0.7 km, representing an average from all available MiniMPL observations."**

Line 227: I suggest rephrasing to: "The goal of optimal estimation is to produce the state vector with maximum a posteriori probability by minimizing the following cost function"

Thanks. Changed as suggested.

Line 346: please explicitly state what the retrieval error is. I would assume "retrieval error" to be the square root of the diagonal elements of the a posteriori covariance matrix, so always positive and smaller than the a priori uncertainties. Explicitly state what the errors are on the horizontal axes of Figs. 10 and 11, are these the % differences between each retrieved quantity and the values used to generate the synthetic spectra?

The errors in Figures 10 and 11 are the retrieval bias (the difference between retrievals and known "truth") from the synthetic experiments. They are different from the "retrieval precision" based on the diagonal elements of the a posteriori covariance matrix. We added these statements in the revised manuscript.

**For each observation scenario in these tests, we calculate the difference between the retrieved state vector and the "truth" that was used to generate the synthetic spectra. The retrieval error (in percentage) is defined as the ratio of the calculated difference to the "truth".**

Line 350: if the a priori aerosol layer height is 700 m with a 50 m uncertainty, an "average error less than 1 km" is large. Does 50 m correspond to the variability in the MiniMPL data? In Fig. 11(c) Why did one of the simulations produce a ~50% error in ALH?

The 50-meter uncertainty is the 1-sigma of the MiniMPL ALH data. From our estimation, the average is about 10%, which is about 70 meters and is actually comparable to the 1-sigma uncertainty.

This outlier with ~50% error is from synthetic experiment "Test 3: Aerosol impact", in which the monthly mean aerosol profile is used to investigate the impact of imperfect aerosol settings, under high AOD (0.1) conditions over the West Pasadena surface site. For this same case, we also observed anomalous an AOD estimate that shows large bias (>60% for AOD=0.1 case, as also shown in Figure 11(b)). We believe that this outlier in ALH and AOD is caused by the bias in the prescribed aerosol composition for a high aerosol loading scenario.

Line 353 and Fig. 12(c): is the "bias In PBL enhancement" the difference between the 3-hourly and monthly a priori profiles? Clarify in the text.

Yes. We added the following statement in the revised manuscript:

**"This bias is defined as the difference in PBL GHG mixing ratios between 3-hourly and monthly a priori atmospheric profiles"**

Line 362: indicate the number of spectra that are included in the analysis after filtering.

We added the following statement in the revised manuscript:

**However, about 20% of the measurements fail to converge, and another 20% fail to pass the post-processing filters; these are discarded. Eventually, 7,733 spectra are available for further analysis.**

Line 375: including a new fitting parameter can only reduce the RMS of spectral fit residuals. This alone is not sufficient to conclude that adding aerosols to the fit improves the retrieval. Did you observe that spectral residual features specifically attributable to the presence of aerosols were reduced?

Yes. The presence of aerosol scattering signal can be clearly seen from the fitting residual of the strong $CO_2$ band. It is known that it is very dark (with very small radiance) over those strong absorption channels. Any extra radiance in these channels is attributable to the contribution from aerosol scattering. Therefore, without aerosol scattering considered, these strong absorption lines show higher residuals, as seen in Fig. A3. These residuals significantly reduced with aerosol scattering contribution considered. We added the following statements in the revised manuscript:

**"For the SCO$_2$ band, since most of the absorption lines are saturated, any extra radiance in this spectral region is attributable to aerosol scattering. Ignoring aerosol scattering results in higher residuals, especially for the strong absorption lines (Appendix Fig. A3). Fitting residuals are significantly reduced using GFIT3."**

Line 391: could you provide a value for the AERONET AOD accuracy?

The uncertainty of aerosol optical depth measured by AERONET is 0.01-0.02 in the 0.34-0.87 μm spectral range, according to Eck et al. (1999). We added this description in the revised manuscript and also added the following reference.

**Eck, T. F., Holben, B. N., Reid, J. S., Dubovik, O., Smirnov, A., O'Neill, N. T., Slutsker, I. and Kinne, S., 1999. Wavelength dependence of the optical depth of biomass burning, urban, and desert dust aerosols. Journal of Geophysical Research: Atmospheres, 104(D24), pp.31333-31349.**

Line 395: is a RMSE value of 0.02 really indicating "good agreement" for a quantity that ranges from ~0.01-0.15? Is it for the coarse or fine AOD, or some combination of both?

This RMSE value is for the total AOD at 1.27 μm, a combination of AODs of coarse- and fine-mode aerosols. Considering the AERONET AOD uncertainty, with is on the order of 0.01-0.02 for 0.34-0.87 μm according to Eck et al. (1999), our estimated RMSE value of 0.02 is very close to the noise level, which indicates good agreement. These statements have been added to Section 5.2 in the revised paper.

Line 403: do we know which other factors?

The other factors may include surface albedo and aerosol properties. For example, a higher aerosol layer has a similar effect on the observed radiance as a combination of darker surface and brighter aerosols (higher SSA). Therefore, the uncertainties from constraining surface reflectance and prescribing aerosol SSA directly influence the ALH estimation.

Line 404-405: In the conclusion you suggest the results could be improved by first retrieving ALH and then using it as a fixed (?) input to retrieve GHGs, this could also be stated here too.

Thanks for the suggestion. We added some sentences in section 5.2 stating this.

Line 430: I found that sentence confusing, specify if you are referring to the O$_2$ or CO$_2$ SCD.

This sentence has been changed to:

**In this case, it is possible for the surface darkening effect to be more dominant than the AOD effect in driving the bias (underestimation) of retrieved XCO$_2$ (or XCH$_4$).**

We have also substantially rephrased this paragraph. Please see Section 6.2 in the revised manuscript.

**Typos:**

Line 45: "Although the GHG retrievals show good agreement with ground-based Total Carbon Column Observing Network (TCCON) results, the retrieved aerosol optical depth (AOD) values have larger differences compared with collocated AERONET measurements"

Check wording (larger => large?), AOD differences with AERONET cannot be "larger" than GHG differences with TCCON.

Thanks for pointing this out. It has been changed to "large".

Line 397: CLARS-FP => CLARS-GFIT or CLARS-FTS?

Corrected. It should be "GFIT3".

Line 421: shorter => shorten

Corrected.